# FlowDec: A flow-based full-band general audio codec with high perceptual quality

**Simon Welker**[♯,*]**, Matthew Le**[◇]**, Ricky T.Q. Chen**[◇]**, Wei-Ning Hsu**[◇]**,
Timo Gerkmann**[♯]**, Alexander Richard**[◇]**, Yi-Chiao Wu**[◇]

[♯]Signal Processing, University of Hamburg, 22527 Hamburg, Germany
[◇]FAIR / Codec Avatar Labs, Meta, 10001 New York / 15222 Pittsburgh, USA

## Abstract

We propose FlowDec, a neural full-band audio codec for general audio sampled at 48 kHz that combines non-adversarial codec training with a stochastic postfilter based on a novel conditional flow matching method. Compared to the prior work ScoreDec which is based on score matching, we generalize from speech to general audio and move from 24 kbit/s to as low as 4 kbit/s, while improving output quality and reducing the required postfilter DNN evaluations from 60 to 6 without any fine-tuning or distillation techniques. We provide theoretical insights and geometric intuitions for our approach in comparison to ScoreDec as well as another recent work that uses flow matching and conduct ablation studies on our proposed components. We show that FlowDec is a competitive alternative to the recent GAN-dominated stream of neural codecs, achieving FAD scores better than those of the established GAN-based codec DAC and listening test scores that are on par, and producing qualitatively more natural reconstructions for speech and harmonic structures in music.

## 1 Introduction

An audio codec is a technique aiming to compress an audio waveform into compact and quantized representations and to reconstruct the audio waveform based on those encoded representations faithfully. The compact and quantized representations are suitable for efficient transmission and storage, which is essential for mobile communications and live video streaming applications (Kroon et al., 1986; Salami et al., 1994; Rao & Hwang, 1996). Different from legacy codecs (Atal & Schroeder, 1970; Schroeder & Atal, 1985; O'Shaughnessy, 1988) which exhibit considerable quality sacrifice in low-bitrate scenarios, modern codecs achieve lossless (Liebchen & Reznik, 2004; Coalson, 2000) or acceptable lossy (Valin et al., 2013; Bessette et al., 2002; Dietz et al., 2015) codings with $2\times$ or $10\times$ compression ratios. However, these codecs usually involve ad hoc designs and extensive manual efforts (Kim & Skoglund, 2024), which hinders the codecs from end-to-end optimizations to achieve high-fidelity audio coding in even lower bitrates (e.g. $<12$ kbit/s).

End-to-end (E2E) Neural codecs (Zeghidour et al., 2021; Défossez et al., 2023; Wu et al., 2023; Kumar et al., 2024) have seen a surge in interest in recent years, particularly due to their usefulness in generative audio tasks such as generating music or speech conditioned on a textual description or transcript. These codecs nowadays achieve very good audio quality at bitrates as low as 8 kbit/s, where most classical non-neural codecs fail to produce acceptable results. To achieve high-quality results at low bitrates, most E2E neural codecs employ adversarial training inspired by generative adversarial networks (GANs) (Goodfellow et al., 2020) to recover natural-sounding signals and to avoid artificial artifacts that arise when training only with waveform or spectral losses.

Score-based (diffusion) and flow-based generative models (Ho et al., 2020; Song et al., 2021; Lipman et al., 2023) have in recent years taken over many generative application domains from GANs. In this spirit, a recently proposed score-based codec is *ScoreDec* (Wu et al., 2024), a widely applicable generative postfilter for E2E neural codecs. ScoreDec aims to recover natural-sounding signals by enhancing codec outputs, removing adversarial losses when training the E2E model and instead

---

*Work completed as part of an internship at Meta

training a *score-based generative model* (Song et al., 2021) as a *postfilter*. While ScoreDec shows a clear advantage in output quality compared to the original codec variants that use adversarial training, it only considers speech signals, was tested only for a relatively high bitrate of 24 kbit/s, and – most importantly – has prohibitively expensive inference at a real-time factor (RTF) of 1.7 caused by the need of around 60 DNN evaluations.

In this work, we propose **FlowDec**, a generative neural codec based on a novel adaptation of conditional flow matching (CFM) (Lipman et al., 2023; Pooladian et al., 2023; Tong et al., 2024), and show that it is a competitive alternative to the current GAN-focused stream of neural codecs for general full-band audio. We address the shortcomings of ScoreDec (Wu et al., 2024) by designing and training for general audio beyond only speech, reducing bitrates from 24 kbit/s to below 8 kbit/s, and reducing the needed number of DNN calls from 60 to 6. We design for full-band audio covering the whole range of human hearing ($\leq 20\,\text{kHz}$) with a 48 kHz sampling rate, to avoid a significant loss of fidelity due to the total removal of high but audible frequencies as in Défossez et al. (2023) or Zeghidour et al. (2021). The key advantage of 48 kHz over 44.1 kHz models such as DAC (Kumar et al., 2024) are that it is easier to achieve whole-number feature rates (75 Hz vs. 86.13 Hz) and bitrates (7500 vs. 7751.95 bit/s) since 48,000 has simpler divisors.

Our main contributions in this work are: **(1)** the extension and simplification of prior score-based generative audio enhancement methods (Welker et al., 2022; Richter et al., 2023; Wu et al., 2024) with a novel adapted CFM method, with theoretical connections and comparisons to recent works on CFM (Pooladian et al., 2023; Tong et al., 2024); **(2)** the application to audio coding and extension of the speech-only ScoreDec (Wu et al., 2024) to general full-band audio at very low bitrates, while reducing the number of DNN evaluations by a factor of 10 without fine-tuning or distillation techniques; **(3)** high-fidelity perceptual quality competitive with a GAN-based state-of-the-art codec (Kumar et al., 2024), which we confirm with objective metrics and listening tests.

## 2 RELATED WORK

### 2.1 NEURAL CODECS

Based on the training objectives, neural audio codecs can be divided into three main categories: auto-encoder (AE), neural vocoder, and postfilter. In the early days, legacy AE-based codecs (Krishnamurthy et al., 1990; Wu et al., 1994; Deng et al., 2010) usually train an AE to reconstruct handcrafted acoustic features and retrieve discrete codes with an independent quantization module on the hidden units which is not globally optimized, and require extensive ad hoc assumptions on audio signals and an additional audio synthesizer. Morishima et al. (1990) propose the first AE speech codec in the waveform domain but do not train the quantizer jointly. The pioneering fully E2E waveform-domain audio codecs incorporate a straight-through gradient estimation (Van Den Oord et al., 2017) or softmax quantization (Kankanahalli, 2018) for joint AE and quantizer training. However, they suffer from either slow inference from autoregressive decoding or limited quality from the lack of effective waveform losses for non-autoregressive (NAR) decoding. Recently, given the significant improvement in NAR audio waveform generation (Yamamoto et al., 2020; Kumar et al., 2019; Kong et al., 2020) adopting GANs (Goodfellow et al., 2020), GAN-based NAR audio codecs (Zeghidour et al., 2021; Défossez et al., 2023; Wu et al., 2023; Kumar et al., 2024) achieve fast coding, impressive audio quality, and low bitrates.

By using the high-fidelity audio generations achieved by neural vocoders (van den Oord et al., 2016; Kalchbrenner et al., 2018; Valin & Skoglund, 2019a; Kong et al., 2020), methods which reconstruct the audio waveform based on quantized handcrafted acoustic features (Klejsa et al., 2019; Valin & Skoglund, 2019b; Mustafa et al., 2021), codes of conventional codecs (Kleijn et al., 2018), or neural AEs (Wu et al., 2023; San Roman et al., 2024), also achieve impressive coding performance. Postfiltering (Zhao et al., 2018; Deng et al., 2020; Biswas & Jia, 2020; Korse et al., 2022) is a similar approach, easing the training burden of abstract code-to-waveform mapping by utilizing the decoder of a pre-trained codec to generate a distorted waveform, which is then enhanced by a postfilter.

### 2.2 SCORE-BASED GENERATIVE SIGNAL ENHANCEMENT

Welker et al. (2022) propose SGMSE, a score-based generative model (SGM) for speech enhancement (SE), by formulating the speech enhancement task as a diffusion process in the complex spectral

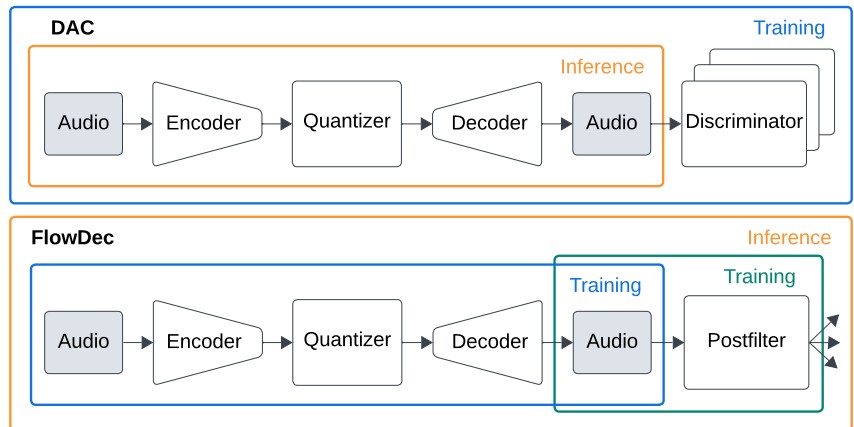

Figure 1: Method overview: Codecs such as DAC (Kumar et al., 2024) employ adversarial training, using multiple specialized discriminator networks trained jointly with the decoder. Our method FlowDec is trained in a non-adversarial two-stage fashion, removing these discriminators and instead adding a stochastic postfilter that can produce multiple enhanced estimates of the pretrained decoder.

domain. To avoid the ad hoc assumption that the additive noise in noisy speech follows a white Gaussian distribution, SGMSE directly incorporates the SE task into the diffusion process by interpolating between clean and noisy spectra, leading to a data-dependent prior similar to PriorGrad (gil Lee et al., 2022). Richter et al. (2023) propose SGMSE+, extending SGMSE to speech dereverberation and significantly improving its quality by using the more powerful backbone NCSN++ (Song et al., 2021) for the score model. Due to the complex spectral modeling, both magnitude and phase spectra are utilized and enhanced, resulting in high-quality speech restoration.

Coding artifacts can also be viewed as a special type of noise that should be removed. To take advantage of both E2E and postfilter approaches, ScoreDec (Wu et al., 2024) adopts SGMSE+ as the postfilter for both conventional and neural codecs and achieves human-level speech quality. However, the inference of ScoreDec is slow due to the high number of diffusion steps, and the effectiveness of ScoreDec for general audio is unclear. To tackle these issues, we propose FlowDec for general audio coding, with significantly reduced runtime cost at a real-time factor below 1, and a simplified formulation that requires only one hyperparameter instead of four.

## 3 METHODS

We cast the problem of recovering an estimate $\hat{x} \in \mathbb{R}^L$ of the clean audio $x^* \in \mathbb{R}^L$ given the code $c := E(x^*)$ from an encoder $E$ as a stochastic inference problem, with the goal of having a model that can provide clean audio estimates $\hat{x}$ as samples from the distribution

$$\hat{x} \sim p_{\text{data}}(\hat{x}|c), \quad c = E(x^*) \in \mathbb{Z}^\ell, \ell \ll L, \tag{1}$$

where $p_{\text{data}}(\cdot|c)$ is the conditional distribution of clean audio given the code $c$. We argue that this treatment is natural, as any encoder $E$ that maps $x^* \in \mathbb{R}^L$ to a lower-dimensional discrete representation $c$ is a many-to-one mapping: multiple $x^*$ will have the same code $c$. Hence, fulfilling the ideal property $D(E(x^*)) = x^*$ is formally impossible if $D$ is a one-to-one mapping. One could instead construct $D$ as an optimal estimator in the mean sense by minimizing

$$\min_D \mathbb{E}_{x^*}\left[\text{dist}(D(E(x^*)), x^*)\right] \tag{2}$$

with a pairwise distance $\text{dist}$ such as the $L^2$ or $L^1$ distance. However, it is known that a method trained this way typically does not produce perceptually pleasing signals (Blau & Michaeli, 2018; 2019) even with domain-specific losses. A popular way around this for neural codecs is to employ adversarial training losses (Zeghidour et al., 2021; Défossez et al., 2023; Kumar et al., 2024) to shift the distribution of decoded signals closer to that of natural signals. While relatively effective, this approach lacks clear interpretability, is limited by the quality of the discriminator, and may fail to properly minimize the distance between $p(\hat{x})$ and $p(x^*)$.

An alternative, which we follow here, is to directly construct $D$ as a one-to-many mapping, as done in recent literature on other audio inverse problems such as speech enhancement, dereverberation, and bandwidth extension (Richter et al., 2023; Lemercier et al., 2023a;b) and most recently for speech coding by Wu et al. (2024). We show a conceptual overview of this idea in Fig. 1. We realize this mapping in the form of a *stochastic decoder* $D_s(c) = \Omega(D_0(c))$, combining a deterministic pre-trained initial decoder $D_0$ with a *stochastic postfilter* $\Omega$. Defining $y := D_0(c)$, $\Omega$ produces conditional samples $\hat{x} \sim p_\Omega(\hat{x}|y)$ from a learned distribution $p_\Omega(\cdot|y)$, which approximates the intractable distribution $p_{\text{data}}(\cdot|y)$ via minimization of a statistical divergence $\mathcal{D}$:

$$p_\Omega = \arg\min_{q_\Omega} \mathcal{D}(q_\Omega(\cdot|y), p_{\text{data}}(\cdot|y)) \tag{3}$$

We can assume that $p_\Omega(\hat{x}|y) = p_\Omega(\hat{x}|c)$ since $D_0$ is known, deterministic and non-compressive. The role of $D_0$ is now to provide a decent initial estimate, which may well still suffer from artifacts and is enhanced by $\Omega$ to deliver perceptually pleasing results. We choose $\mathcal{D}$ as the Wasserstein-2 distance, and practically minimize equation 3 by training a *flow model*, a neural network $v_\theta$ trained with an adapted CFM objective (Lipman et al., 2023).

## 3.1 Flow Matching

Lipman et al. (2023) introduce the idea of Flow Matching, where the goal is to learn a model that can transport samples from a tractable distribution $q_0(x_0)$ to an intractable data distribution $q_1(x_1) = p_{\text{data}}$ by solving the neural ordinary differential equation (ODE)

$$\frac{d}{dt}\phi_t(x) = u_t(\phi_t(x)), \quad \phi_0(x) = x_0 \tag{4}$$

starting from a sample $x_0 \sim q_0$. We call $\phi_t : [0,1] \times \mathbb{R}^N \to \mathbb{R}^N$ the *flow* with the associated *time-dependent vector field* $u_t : [0,1] \times \mathbb{R}^N \to \mathbb{R}^N$, which generates a *probability density path* $p_t : \mathbb{R}^N \to \mathbb{R}_{>0}$ with $p_{t=0} = q_0$ and $p_{t=1} = q_1$. They propose to learn $v_\theta$ with the CFM target:

$$\mathcal{L}_{\text{CFM}} := \mathbb{E}_{x,t,p_t(x|x_1)} \left[ \|v_\theta(x,t) - u_t(x|x_1)\|_2^2 \right] \tag{5}$$

where $x_1 \sim q_1$ and $\mathcal{L}$ denotes a training loss function. A key insight is that the *conditional* Eq. (5) has the same gradients as an intractable *unconditional* flow matching objective (Lipman et al., 2023, Eq. 5), and marginalizes to the correct unconditional probability path $p_t(x)$ and flow field $u_t(x)$.

## 3.2 Joint Flow Matching for Signal Enhancement

In the original flow matching (Lipman et al., 2023) and score matching (Song et al., 2021) formulations, $x_0{}^1$ is sampled independently of $x_1$, typically from a zero-mean Gaussian $q_0 = \mathcal{N}(0, \sigma^2 I)$. Pooladian et al. (2023) and Tong et al. (2024) show that, while the *conditional* paths $p_t(x|x_1)$ fulfill optimal transport (OT) from $q_0$ to $q_1$ when $q_0$ is a standard Gaussian, the modeled *marginal* probability path $p_t(x)$ generally does not fulfill OT. This can lead to high-variance training and low straightness in the learned marginal flow field $v_\theta$, and thus to inefficient inference and suboptimal sample quality. To rectify this, both works propose a per-batch approximation to OT between the full distributions, by reordering the pairings in each training batch $\{(x_{b,0}, x_{b,1})\}_{b=1}^B$ with *optimal couplings* determined by an OT algorithm on each batch. Effectively, this samples $(x_0, x_1) \sim q(x_0, x_1)$ *jointly* rather than independently.

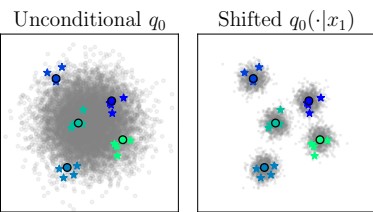

Figure 2: Unconditional $q_0(x_0)$ versus our $q_0(x_0|x_1)$. Colored dots represent $y$, stars are associated $x^*$.

Here, we also propose sampling $(x_0, x_1)$ jointly, but in a way that is adapted to enhancement tasks and does not require any OT solvers or extra computations. Concretely, since we have access to the initial estimate $y = D_0(c) = D_0(E(x^*))$, we choose the probability path

$$p_t(x_t|x_1, y) = \mathcal{N}(x_t; \mu_t, \sigma_t) := \mathcal{N}(x_t; y + t(x_1 - y), (1-t)^2 \Sigma_y) \tag{6}$$

---

[1] Note the different notational convention in score-based works, where the meaning of $x_0$ and $x_1$ is reversed.

where $\Sigma_y = \text{diag}(\sigma_y^2)$ is a diagonal covariance matrix. This probability path is a linear interpolation between $y$ and $x_1$, with noise linearly decreasing from $\sigma_y$ to zero. This leads to a coupling between $x_0$ and $x_1$ through $y$. Namely, $q_0(x_0|x_1, y) = \mathcal{N}(x; y, \Sigma_y)$, i.e., the mean of $x_0$ is shifted from 0 to $y$, similar to score-based signal enhancement works (Richter et al., 2023; Wu et al., 2024). Intuitively, while we do not use it for inference or training, the marginalized $q_0(x_0)$ is now a mixture of Gaussians, each of variance $\sigma_y^2$ and centered at the respective $y$ from the training data, see Fig. 2. When $\sigma_y$ is well-chosen so these Gaussians have negligible overlap, no minibatch OT is needed as the per-batch couplings can be assumed optimal by construction. We find that the choice of $\sigma_y$ is important for output quality, see Appendix A.1 for more details. The conditional $u_t$ can be found via (Lipman et al., 2023, Eq. 15), with the full derivation in Appendix A.2:

$$u_t(x|x_1, y) = \frac{x_1 - x_t}{1 - t} \tag{7}$$

To simplify, since $x_t$ can be written in terms of $x_0$, we note that

$$\begin{align}
x_t &= tx_1 + (1-t)x_0, & x_0 &\sim \mathcal{N}(x_0; y, \Sigma_y)) \tag{8} \\
&= tx_1 + (1-t)y + (1-t)\sigma_t\varepsilon, & \varepsilon &\sim \mathcal{N}(0, I) \tag{9} \\
x_0 &= y + \sigma_y\varepsilon, & \varepsilon &\sim \mathcal{N}(0, I) \tag{10}
\end{align}$$

which, using that $x_1 = x^*$ from Eq. (6), leads to the simple joint flow matching loss

$$\mathcal{L}_{\text{JFM}} := \mathbb{E}_{t \sim \mathcal{U}(0,1), (x^*, y) \sim \mathfrak{D}, \varepsilon \sim \mathcal{N}(0,I), x_t \sim p_t(x_t|x_0)} \left[ \left\| v_\theta(x_t, t, y) - (\underbrace{x^*}_{=x_1} - \underbrace{(y + \sigma_y\varepsilon)}_{=x_0}) \right\|_2^2 \right] \tag{11}$$

where $\mathfrak{D}$ is the training dataset. Note also that this loss removes the numerical instability around $t \approx 1$ of Eq. (7) by reparameterizing in terms of $x_0$ and $x_1$. By choosing $\sigma_y > 0$, we enforce the flow field to be a contractive mapping. This ensures the ODE for inference is numerically stable and converges locally. Our choice of $p_t$ improves upon SGMSE (Welker et al., 2022; Richter et al., 2023; Wu et al., 2024), in that trajectories in our formulation can reach $x^*$ exactly, which SGMSE fails to do since it does not model the correct $q_0$ (Lay et al., 2023). We also avoid designing and tuning special stochastic differential equations (SDEs) with multiple hyperparameters and use only one hyperparameter, $\sigma_y$, for which we propose a data-based heuristic

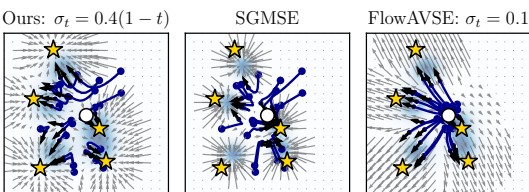

Figure 3: Flow field comparison at $t = 0.7$ for our linear $\sigma_t$ (left) versus score-based SGMSE (center) and FlowAVSE with constant $\sigma_t$ (right) for a toy problem. The white dot is $y$, yellow stars are possible $x^*$, blue lines are sample trajectories, and the background color indicates the density $p_t$. SGMSE has highly curved trajectories and does not contract to $x^*$; FlowAVSE is non-contractive.

(Appendix A.1). Another recent work for audiovisual speech enhancement by Jung et al. (2024) also makes use of CFM, but uses an independent CFM formulation (Tong et al., 2024) resulting in a constant $\sigma_t = \sigma$ and the target flow field $u_t$ being independent of the sampled noise. This leads to a non-contractive flow field and the potential for residual noise being left in the estimates since $\sigma_1 = \sigma > 0$, whereas $\sigma_1 = 0$ in our case. We illustrate this qualitatively in Fig. 3 and also show empirically in our results section that, for our postfiltering task, our formulation leads to better quality than both alternatives, at both a low and high number of function evaluations (NFE).

In practice, we replace $x^*, y$ with the feature representations $\mathbf{X}^*, \mathbf{Y}$ from an invertible feature extractor $\Phi$ and learn the flow in this feature domain. Namely, $\Phi$ is an amplitude-compressed complex short-time Fourier transform (STFT) (Welker et al., 2022) with compression exponent $\alpha = 0.3$, see Appendix A.4 for details. We provide $\mathbf{Y}$ to $v_\theta$ as conditioning via channel-wise concatenation at the input (Richter et al., 2023).

After training, the flow model $v_\theta$ together with the ODE (4) models the conditional distribution $p_\Omega(\mathbf{X}^*|\mathbf{Y})$. To produce clean feature estimates $\hat{\mathbf{X}} \sim p_\Omega$, we first sample an initial state (latent) $\mathbf{X}_0 \sim q_0(\mathbf{X}_0|\mathbf{Y})$ and then solve the flow ODE (4) using $v_\theta$ from $t = 0$ to $t = 1$ with a numerical ODE solver to get $\hat{\mathbf{X}}_1$. We use the Midpoint solver with 3 steps (NFE = 6) unless otherwise noted, due to its improved quality over the Euler solver at a low NFE, see Appendix A.7.6. Finally, we apply the inverse of the feature extractor $\Phi$ to produce the waveform estimate $\hat{x} = \Phi^{-1}(\hat{\mathbf{X}}_1)$.

### 3.3 Non-adversarial Codec Training

Due to a lack of effective phase losses, NAR audio generative models trained with only spectral losses usually exhibit buzzy noise caused by unsynchronized phases. Many works employ adversarial training to circumvent this and restore more natural-sounding audio. This however requires complex handcrafted multi-discriminator losses and weightings to avoid unstable training, mode collapse, and divergence, and lacks interpretability (Wu et al., 2024; Lee et al., 2024). In recent years, generative diffusion models have largely superseded GANs for image and audio generation due to easier training and better detail modeling (Dhariwal & Nichol, 2021), but have yet to make such a strong impact for audio codecs.

To overcome these issues, we remove adversarial training and instead use a generative postfilter. We train a deterministic neural codec as the initial decoder $D_0$ without any adversarial losses and leave the task of matching the distributions of output audio and clean audio to the stochastic postfilter $\Omega$. The simplest way forward, which we follow, is to take an existing state-of-the-art neural codec such as DAC (Kumar et al., 2024) as $D_0$ and to remove all components related to adversarial loss terms.

### 3.4 Underlying Codec: Improved non-adversarial DAC

In principle, stochastic postfilters such as ours can be trained for any underlying codec to enhance its waveform estimates, as shown in Wu et al. (2024). We use DAC (Kumar et al., 2024) as the basis for our underlying codecs due to its status as a state-of-the-art neural codec, as also recently established for speech by Muller et al. (2024), and its adaptibility for other sampling rates and bitrates. We remove the adversarial losses and modify some configuration settings listed in Section 4.2.

When we first trained this non-adversarial codec, we found that it produced unnatural results and bad scale-invariant signal-to-distortion ratio (SI-SDR) values around -30 dB, particularly for music. After finding that low frequencies ($\leq 2\,\text{kHz}$) were badly modeled we add a multiscale constant-Q transform (CQT) loss, inspired by the high low-frequency resolution of the CQT, frequent use of the CQT in music processing (Moliner et al., 2023), and the multiscale Mel losses used by DAC. As in DAC's multiscale Mel loss, we use both the differences of amplitudes and of log-amplitudes. We further add a $L^1$ waveform-domain loss to improve SI-SDR values and phase errors that magnitude-only losses are blind to. We demonstrate the effectiveness of these losses in Appendix A.7.2.

### 3.5 Frequency-dependent Noise Levels

As noted in Section 3.1, the choice of $\sigma_y$ is important for output quality. It is well known that the power spectrum of most natural signals follows an inverse power law, so high frequencies have much lower power than low frequencies. A single scalar $\sigma_y$ can thus potentially lead to oversmoothing when the added Gaussian noise dominates high frequencies, as also previously observed for images (Kingma & Gao, 2023, Appendix J). To rectify this, we calculate frequency-dependent curves $\sigma_y(f)$ by performing the heuristic quantile calculation in Eq. (12) independently for each STFT frequency band. Similarly, MBD (San Roman et al., 2023) proposes a band-dependent noise scale but uses only 4 broad Mel bands for this purpose. We demonstrate the effectiveness in Appendix A.7.4.

## 4 Experimental Setup

### 4.1 Datasets

For **underlying codec training**, we prepare a varied combination of datasets containing music, speech, and sounds, which are listed in Table 1. As proposed in Kumar et al. (2024), we sample audios in a type-balanced way during training, i.e., each training batch contains – in expectation – the same number of speech files as music files and sound files.

For **postfilter training**, we use the same overall dataset as a basis but perform the following additional steps: **(1)** To avoid slow postfilter training from calling $D_0$ in every step, we randomly sample 100,000 clean files $x^*$ per audio type and crop out segments with a maximum 30-second duration, calculate $y = D_0(E(x^*))$, and store it on disk. **(2)** For the postfilter to learn complex audio scenarios, we increase data variety with 100,000 clean 10-second mixtures of all three audio types from the

Table 1: Datasets used for codec training. Datasets in [brackets] are internal. $f_s^{\max}$ denotes the maximum sampling frequency and "h" is short for hours. For WavCaps-FreeSound*, we filter the part of FreeSound contained in WavCaps to keep only the files with commercial-friendly licenses. For CommonVoice 13.0* we use a custom subset.

| Dataset | Duration | $f_s^{\max}$ | Type |
|---|---|---|---|
| MSP-Podcast (Lotfian & Busso, 2019) | 103 h | 16 kHz | Speech |
| CommonVoice 13.0* (Ardila et al., 2020) | 1602 h | 16 kHz | Speech |
| LibriTTS (Zen et al., 2019) | 553 h | 24 kHz | Speech |
| EARS (Richter et al., 2023) | 100 h | 48 kHz | Speech |
| VCTK 84spk (Valentini-Botinhao, 2017) | 20 h | 48 kHz | Speech |
| LibriVox (Kearns, 2014) | 55611 h | 16 kHz | Speech |
| Expresso (Nguyen et al., 2023) | 20 h | 48 kHz | Speech |
| [InternalSpeech] | 1512 h | 48 kHz | Speech |
| [InternalMusic] | 18949 h | 32 kHz | Music |
| WavCaps-FreeSound* (Mei et al., 2024) | 1582 h | 32 kHz | Sound |
| [InternalSound] | 5309 h | 48 kHz | Sound |

subsets described above. We mix each randomly paired three audios in random proportions with mixing coefficients $(w_{\text{speech},k}, w_{\text{music},k}, w_{\text{sound},k})$ sampled from a Dirichlet distribution $\text{Dir}(\alpha_{\text{speech}} = 4, \alpha_{\text{music}} = 2, \alpha_{\text{audio}} = 1)$. We repeat all constituent segments shorter than 10 seconds and center-crop all that are longer. This leaves us with 400,000 pairs (2778 hours) of data.

As our **test set**, we use 3,000 random audio samples with 1,000 of each audio type: 500 files from the VCTK test set (Valentini-Botinhao, 2017) and 500 from the EARS test set (Richter et al., 2024b) for speech, 500 files from MUSDB18-HQ (Rafii et al., 2019) and 500 from MusicCaps (Agostinelli et al., 2023) for music, and 1000 files from AudioSet (Gemmeke et al., 2017) for sound. To avoid overlap with MusicCaps, we remove all files from AudioSet with music-related tags, but keep tags related to instruments. We crop audios to a 10-second duration. As MUSDB, MusicCaps, and AudioSet are not used for training, we sample from them without regard to train/test splits.

## 4.2 MODEL TRAINING AND VARIANTS

For our **underlying codecs**, we use the official code and training settings from DAC (Kumar et al., 2024) but remove adversarial losses (Section 3.3), add a CQT and waveform loss (Section 3.4), and modify the configuration as listed in Table 2. We call these underlying codecs *NDAC* to avoid confusion with the adversarially trained DAC. NDAC-75 is targeted at 48 kHz audio with a whole-number feature rate (75 Hz) and whole-number bitrates. NDAC-25 is a variant tailored for downstream generative audio tasks, with

Table 2: Our underlying codec variants, compared to official 44.1 kHz DAC by Kumar et al. (2024). $f_s$ is the sampling rate in kHz, $H$ is the hop length in samples, $f_{\text{feat}}$ is the feature rate in Hz, $n_c$ is the number of codebooks, and $d_{\text{emb}}$ is the latent code embedding dimension. Bitrates are in kbit/s.

| Name | Bitrates | $f_s$ | $H$ | $n_c$ | $d_{\text{emb}}$ |
|---|---|---|---|---|---|
| DAC | 0.86–7.75 | 44.1 | 512 | 9 | 1024 |
| NDAC-75 | 0.75–7.50 | 48 | 640 | 10 | 1024 |
| NDAC-25 | 0.25–4.00 | 48 | 1920 | 16 | 128 |

a lower feature rate (25 Hz) and feature dimension which are advantageous for audio generation due to more efficient memory usage and decreased modeling difficulties. For the CQT loss (Section 3.4), we use the `CQT2010v2` implementation of the CQT from the `nnAudio` Python package with 9 octaves, hop length 256, minimum frequency 27.5 Hz, $\{16, 32, 48, 64, 80\}$ bins per octave, with a loss weight of 1 for music samples and 0 for audio and speech samples. For the $L^1$ waveform loss, we use a weight of 50. We train for 800,000 iterations with 0.4 second snippets and a batch size of 72. As baselines, we train *DAC-75* and *DAC-25*, equivalent versions of NDAC-75 and NDAC-25 with the original adversarial losses. To show that the differences between FlowDec and DAC are not just caused by the extra parameters from the postfilter, we also train baselines *2xDAC-75* and *2xDAC-25* for which we double the channels of all decoder convolution layers, increasing the parameters by +100 M vs. +26 M from the postfilter.

As **postfilters**, we train the following variants based on NDAC-75 and NDAC-25:

1. *FlowDec-75m*: 75 Hz, multi-bitrate. Trained based on NDAC-75 with bitrates {7.5, 6.0, 4.5, 3.0} kbit/s, by setting the number of codebooks at inference to {10, 8, 6, 4}. We include only this set of bitrates for ease and speed of training, and because we found that the codec does not provide good results below 3.0 kbit/s.

2. *FlowDec-75s*: 75 Hz, single-bitrate. Trained based on NDAC-75 using only the highest bitrate of 7.5 kbit/s. The goal of this variant is to serve as a baseline for ablations and to investigate the quality gap between a single- and multi-bitrate postfilter.

3. *FlowDec-25s*: 25 Hz, single-bitrate. Trained based on NDAC-25 with a bitrate of 4.0 kbit/s. We do not train for multiple bitrates here as the bitrate and feature rate is already very low.

We train all postfilters based on a slightly modified NCSN++ architecture (Song et al., 2021) with 26 M parameters (details in Appendix A.3). We use Adam (Kingma, 2014) at a learning rate of $10^{-4}$ for 800,000 iterations, a 2-second snippet duration, and a batch size of 64. We track an exponential moving average (EMA) of the weights with decay 0.999 for inference. For every variant, we train one version with global $\sigma_y$ and one with a frequency-dependent $\sigma_y(f)$, see Section 3.5. We use the frequency-dependent variants for all results unless stated otherwise. For the global variants, we set $\sigma_y = 0.66$. For the frequency-dependent variants, we estimate 768-point frequency curves $\sigma_y(f)$ and smooth them with a Gaussian kernel of bandwidth 3. We train further models for ablation studies (Appendix A.7) based on FlowDec-75s.

### 4.3 OBJECTIVE METRIC EVALUATION

For evaluation with objective metrics we use SI-SDR (Roux et al., 2019), Frechét Audio Distance (FAD) with `clap-laion-audio` embeddings as proposed in Gui et al. (2024), frequency-weighted segmental signal-to-noise-ratio (fwSSNR) (Loizou, 2013), the neural ITU-T P.804 estimation method SIGMOS (Ristea et al., 2024), and logSpecMSE, i.e., the mean squared error (MSE) of decibel log-magnitude spectrograms with a 32 ms Hann window and 75% overlap. Note that SIGMOS is only valid for speech signals, so we only evaluate it on the speech test audios.

### 4.4 SUBJECTIVE LISTENING TESTS

Since objective metrics generally do not tell the full story of how a method is perceived by human listeners (Torcoli et al., 2021), it is important to also test this perceived quality directly. We conduct two MUSHRA-like tests (ITU, 2015) detailed in Table 3, comparing FlowDec variants against their DAC equivalents. "Test A" is designed to test our main models, and "Test B" to test low feature rate (25 Hz) models. We use Opus (Valin et al., 2013)

Table 3: Listening test parameters. Bold numbers in parentheses denote the bitrates in kbit/s.

| Test | Compared methods |
| --- | --- |
| A | FlowDec-75m **(7.5, 4.5)**, FlowDec-75s **(7.5)**, DAC-75 **(7.5, 4.5)**, EnCodec **(6.0)**, Opus **(7.5)** |
| B | FlowDec-25s **(4.0)**, FlowDec-75m **(4.5)**, DAC-25 **(4.0)**, DAC-75 **(4.5)**, Opus **(4.0)** |

at the highest used bitrate as the low anchor and include the original audio as the hidden reference. In Test A, we also include the official 48 kHz checkpoint of EnCodec (Défossez et al., 2023) at 6.0 kbit/s for comparison. We conduct both tests with 21 random 10-second audios from our test set: 7 from the EARS test set, 7 from MUSDB, and 7 from AudioSet. We ask 15 expert listeners to rate each audio on a scale from 0 to 100. We exclude listeners that rated the reference $< 90$ or the low anchor $> 90$ for more than 15% of trials, resulting in 11 listeners for Test A and 10 for Test B.

## 5 RESULTS

### 5.1 OBJECTIVE METRICS

In Fig. 4, we show the objective metric results of FlowDec-75m and FlowDec-75s compared to EnCodec (48 kHz), DAC-75, 2xDAC-75 and the official DAC 44.1 kHz checkpoint, and also include

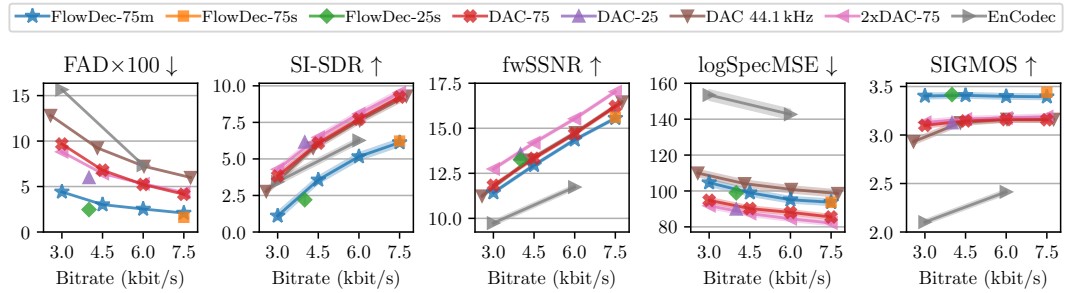

Figure 4: Mean objective metrics attained by compared methods on the test set at varying bitrates. Colored bands indicate 95% confidence intervals. SIGMOS is speech-only and is calculated only on the speech test files. FAD is multiplied by 100 for readability. Numbers can be found in Table 8.

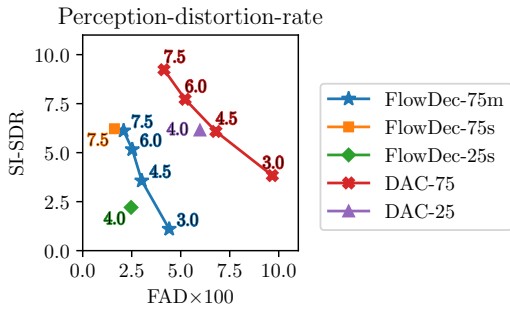

Figure 5: Perception (FAD) – distortion (SI-SDR) – rate tradeoff (Blau & Michaeli, 2019) of compared methods. Numbers next to points indicate the bitrate in kbit/s.

Table 4: FAD×100, mean SI-SDR, and mean fwSSNR of FlowDec-75s versus the related Score-Dec (Wu et al., 2024) and constant $\sigma_t$ (Jung et al., 2024). Best in bold.

| Method | FAD×100 | SI-SDR | fwSSNR |
|---|---|---|---|
| **NFE = 6** | | | |
| FlowDec | **1.62** | 7.55 | **15.46** |
| ScoreDec | 145.30 | -27.23 | 3.15 |
| $\sigma_t = 0.05$ | 28.88 | 9.95 | 5.50 |
| $\sigma_t = 0.66$ | 29.83 | **10.10** | 6.55 |
| **NFE = 50** | | | |
| FlowDec | **1.34** | 7.41 | **15.65** |
| ScoreDec | 5.73 | **7.50** | 14.45 |

the 25 Hz feature rate models FlowDec-25s and DAC-25 for comparison. Our main model FlowDec-75m produces the best FAD values by a large margin and also performs best on the SIGMOS OVRL metric. For the intrusive spectral metrics SI-SDR, fwSSNR, and logSpecMSE, retrained DAC generally outperforms FlowDec, though the gap in the perceptually weighted fwSSNR is small. This is to be expected under the *perception-distortion tradeoff* discussed in (Blau & Michaeli, 2018; 2019): FlowDec favors better perception (FAD) along this tradeoff at the cost of increased distortion (SI-SDR), see also Fig. 5, similar to observations made about score-based models for speech enhancement (Richter et al., 2023) and JPEG artifact removal (Welker et al., 2024). Furthermore, we see that the single-bitrate FlowDec-75s slightly outperforms FlowDec-75m at 7.5 kbit/s as expected, and that 2xDAC is slightly better than DAC but does not fundamentally change the qualitative behavior of DAC. For the 25 Hz models, we can see that the general behavior of FlowDec and DAC is unchanged, with FlowDec again exhibiting better FAD and SIGMOS.

In Table 4, we compare FAD, SI-SDR and fwSSNR of FlowDec-75s at NFE $\in \{6, 50\}$ against ScoreDec (Wu et al., 2024) and the alternative flow-based formulation with constant $\sigma_t$ (Jung et al., 2024). We can see that for NFE = 6, FlowDec is a clear improvement over ScoreDec which produces unusable results at this NFE and also performs significantly better than Jung et al. (2024) here. At NFE = 50, ScoreDec and FlowDec achieve similar SI-SDR, but FlowDec performs significantly better in FAD. A full metric comparison table can be found in Appendix A.7.1. Finally, in Fig. 7, we show a qualitative spectrogram comparison of FlowDec compared DAC for a guitar recording, which illustrates better reconstruction of harmonic structures by FlowDec. We show more example spectrogram comparisons, including the worst reconstructions from FlowDec, in Appendix A.8.

## 5.2 SUBJECTIVE LISTENING TESTS

In Fig. 6, we show the results from both subjective listening Tests, A and B, as boxplots of MUSHRA scores per method and bitrate. For Test A, we can see that the 4.5 kbit/s variants are rated somewhat lower than the 7.5 kbit/s variants but still achieve good scores compared to EnCodec at 6.0 kbit/s, and

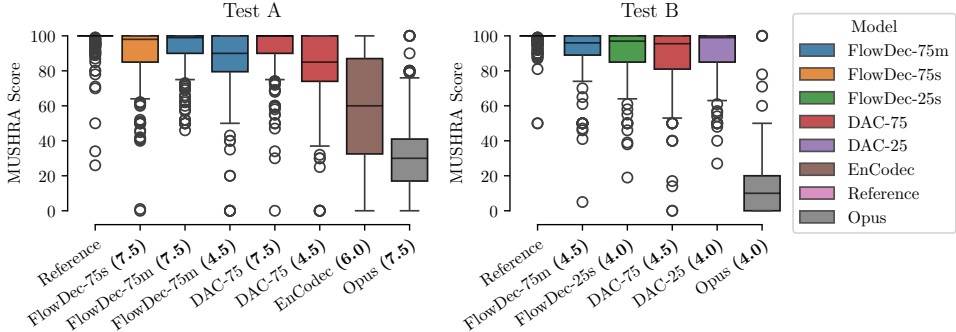

Figure 6: Subjective listening results from Test A (left) and Test B (right). Numbers in (parentheses) denote the used bitrate in kbit/s. FlowDec is rated on par with DAC (Kumar et al., 2024), with no significant differences between their score distributions at any given bitrate and feature rate.

the low anchor Opus. We can further see that, at any given bitrate, the score distributions of DAC-75 and FlowDec-75m show no significant differences. For Test B with the 25 Hz models, we can again see that DAC and FlowDec generally perform on par, and also that the 25 Hz models are rated very similarly as their higher feature rate (75 Hz) equivalents at a similar bitrate. In Appendix A.6, we also show results split by audio type, which seem to suggest that FlowDec performs better than DAC for speech samples, slightly worse for sound samples, and on par for music.

## 5.3 REAL-TIME FACTOR

An important property of a codec is its runtime. We determine the real-time factor (RTF) of the two NDAC variants and the FlowDec postfilter at NFE $\in \{4, 6, 8\}$ with the midpoint solver on an NVIDIA A100-SXM4-80GB GPU. We find an RTF of 0.0134 for NDAC-75 and 0.0084 for NDAC-25. For the postfilter, we find that RTF $\approx 0.0358 \times$ NFE. At our default setting NFE $= 6$, this results in a total RTF of **0.2285** for FlowDec-75(m/s) and **0.2235** for FlowDec-25s, a significant improvement over the RTF of 1.707 for ScoreDec (Wu et al., 2024).

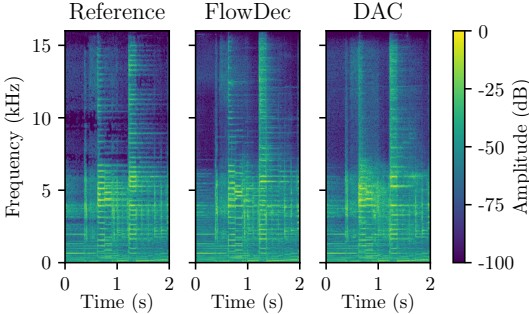

Figure 7: Spectrogram comparison (pre-emphasis of 0.95) of DAC and FlowDec at 7.5 kbit/s for a guitar test audio. FlowDec better preserves harmonics where DAC creates noise-like structures.

## 6 CONCLUSION

We presented FlowDec, a novel postfilter-based neural codec for general audio with high perceptual quality. FlowDec uses a novel modification of the flow matching formalism for signal enhancement, which is inspired by previous score- and flow-based generative works for signal enhancement (Richter et al., 2023; Wu et al., 2024; Jung et al., 2024) but improves upon them both in terms of theoretical properties and output quality. We showed that FlowDec achieves state-of-the-art FAD scores for the coding task and, in a listening test, performs on par with the current state-of-the-art GAN-based codec DAC (Kumar et al., 2024) at bitrates between 4.5 and 7.5 kbit/s. Furthermore, FlowDec also shows promising quality at the very low feature rate of 25 Hz and bitrate of 4.0 kbit/s, which we hope can contribute to more efficient long-range generative audio modeling.

While FlowDec, like DAC, is currently not streaming-capable due to the noncausal architecture of the used DNNs, our postfilter approach can be modified for a causal DNN as in (Richter et al., 2024a), which would pave the way for real-time communication and audio streaming applications. We leave this for future work, particularly since there are currently no streaming codecs available that achieve the quality of DAC to our knowledge. Another interesting future direction is the joint training of the initial decoder and the postfilter similar to Lemercier et al. (2023b), which could improve quality but may lead to unstable training. Finally, as the NCSN++ architecture we use was originally built for images, we expect that future work using DNN architectures better adapted to audio signals can further improve the quality of FlowDec.

## REPRODUCIBILITY STATEMENT

To ensure the reproducibility of our work, we provide all necessary details on the mathematical formulations and loss functions in Section 3.1 and Appendix A.2, of the network architecture in Appendix A.3, and the feature representation in Appendix A.4. We explicitly describe the proposed modifications to non-adversarial DAC in Section 3.4, and provide all hyperparameters for this modification along with the training details of both this underlying codec and all postfilters in Section 4.2. We provide the full list and details for all datasets, besides internal datasets which at present cannot be open-sourced, in Section 4.1. We note that we used only a small fraction of this total training data for training our FlowDec postfilter, with most being used for training the underlying codecs (see Section 4.1). Our underlying codecs are based on DAC (Kumar et al., 2024) and can straightforwardly be retrained with the public datasets listed in their work, using their available codebase, and the additional implementation details for our proposed CQT loss listed in Section 3.4. To further ensure reproducibility, we have open-sourced our code for FlowDec training and inference, along with pretrained model checkpoints of the FlowDec models listed in this paper, made available at `https://github.com/facebookresearch/FlowDec`. A demo page is available at `https://sp-uhh.github.io/FlowDec/`.

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

## A    APPENDIX

### A.1    A HEURISTIC FOR CHOOSING $\sigma_y$

An important question arising in Section 3.1 is how to set $\sigma_y$. In Fig. 8, we show the effects of three $\sigma_y$ settings on a simple toy problem. A too-large $\sigma_y$ leads to a regression to the mean effect, pointing the flow field towards the mean of all viable clean audios $x^*$ for most times $t$, which results in oversmoothing and bad perceptual quality. On the other hand, a very small $\sigma_y$ close to 0 does not allow learning the flow field well, as most regions of the space have very low probability during training. Similar to the visually guided setting $\sigma_y = 0.4$ in Fig. 8, we find that the following heuristic works well for all of our cases:

$$\sigma_y = \frac{1}{3}\sqrt{Q(|\mathbf{X}^* - \mathbf{Y}|^2, 0.997)} \tag{12}$$

where $Q$ is the quantile operation. Similarly to a root mean squared error (RMSE), this is the *root of the 0.997th quantile of squared errors* induced by the initial decoder $D_0$ in the feature domain. The constants $\frac{1}{3}$ and 0.997 are inspired by the 3-sigma rule of a Gaussian distribution. The chosen $\sigma_y$, 0.66 for our FlowDec models, then covers all viable estimates $\mathbf{X}^*$, except outliers beyond the 0.997th quantile, within the 3-sigma region of the added Gaussian noise around $\mathbf{Y}$.

### A.2    DERIVATION OF CONDITIONAL FLOW FIELD

Referring to Section 3.2, we perform the derivation of the target flow field $u_t$ in more detail here. We can find the target flow field $u_t$ from our chosen probability path, $p_t$ Eq. (6), using (Lipman et al., 2023, Eq. 15):

$$u_t(x|x_1, y) = \frac{\sigma_t'(x_1, y)}{\sigma_t(x_1, y)}(x - \mu_t(x_1, y)) + \mu_t'(x_1, y) \tag{13}$$

$$= \frac{-\sigma_y}{(1-t)\sigma_y}(x_t - (y + t(x_1 - y))) + (x_1 - y) \tag{14}$$

$$= -\frac{x_t - y - tx_1 + ty}{1-t} + \frac{(1-t)(x_1 - y)}{1-t} \tag{15}$$

$$= \frac{-x_t + y + tx_1 - ty + x_1 - tx_1 - y + ty}{1-t} \tag{16}$$

$$= \frac{x_1 - x_t}{1-t} \tag{17}$$

$$\tag{18}$$

which matches the expression (Lipman et al., 2023, Eq. 21) of the flow field for an unconditional zero-mean $x_0$ when their $\sigma_{\min} = 0$. We can further see that

$$u_t(x|x_1, y) = \frac{x_1 - x_t}{1-t} \tag{19}$$

$$= \frac{x_1 - (tx_1 + (1-t)x_0)}{1-t} \tag{20}$$

$$= \frac{(1-t)x_1 - (1-t)x_0}{1-t} \tag{21}$$

$$= x_1 - x_0 \tag{22}$$

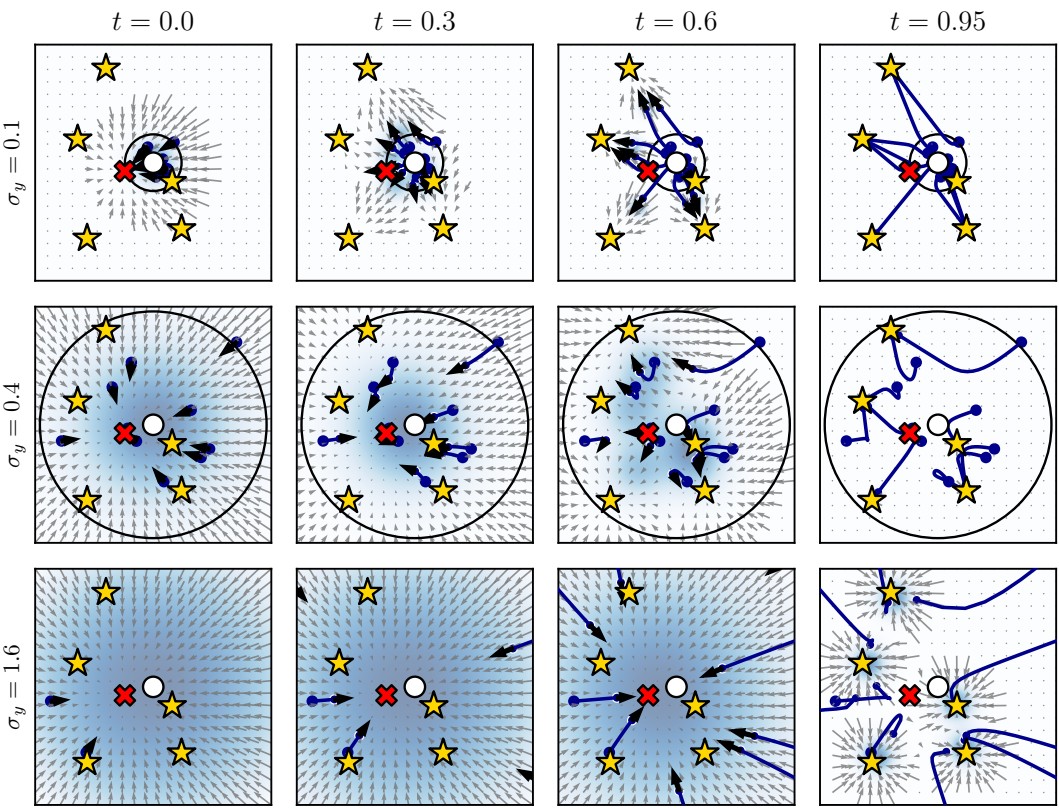

Figure 8: Flow fields for our FlowDec formulation at different times $t$ and settings of the hyperparameter $\sigma_y$, illustrated on a toy problem. The white dot represents the initial estimate $y$, the yellow stars represent possible target signals $x^*$, and the red cross is the mean of all $x^*$. The background shows the probability density $p_t$ and the circle indicates $3\sigma_y$ around $y$. The flow field for large $\sigma_y = 1.6$ points towards the mean (red cross) for most $t$, while for $\sigma_y = 0.4$, it points towards each viable point much earlier. While a low $\sigma_y = 0.1$ leads to the straightest paths, it also results in most regions of the space having very low probability $p_t$ for all $t$ of being sampled during training, which is problematic under model and truncation errors since it is much more likely that trajectories fall off the small high-probability regions.

where $x_1 = x^*$ is exactly the target clean signal $x^*$ since $\sigma_1 = 0$, and $x_0 \sim \mathcal{N}(x_0; y, \Sigma_y)$ is a sample from a Gaussian with mean $y$ (the initial decoder output) and diagonal covariance $\Sigma_y = \mathrm{diag}(\sigma_y^2)$. Hence, we can reparameterize $x_0$ as

$$x_0 = y + \sigma_y \varepsilon, \qquad \varepsilon \sim \mathcal{N}(0, I) \tag{23}$$

which, together with Eq. (22) and $x_1 = x^*$, leads exactly to the expression used in our loss, Eq. (11).

### A.3    NCSN++ NEURAL NETWORK CONFIGURATION DETAILS

For our postfilter flow model, we reconfigure the NCSN++ 2-D U-Net architecture (Song et al., 2021) used in prior audio works (Richter et al., 2023; Wu et al., 2024). In preliminary investigations, we found that the original architecture can produce high-frequency harmonic artifacts in music, see Appendix A.7.5. We found that doubling the channels ($128 \to 256$) at the first two U-Net depths effectively suppresses these artifacts. An explanation may be that the capacity of only 128 filters in the early layers may not be enough for the increased sampling rate and data complexity (speech $\to$ music, sound, speech) compared to Richter et al. (2023). To counteract the increased memory usage, we reduce the depth from 7 to 4 and reduce the channels at depths 3 and 4 from 256 to 128. We use 1 instead of 2 ResNet blocks per depth as in (Lemercier et al., 2023b). Finally, we remove all attention-based layers to ensure that the inference runtime is linear in the audio duration. Our architecture has 26 M parameters instead of the original 65 M.

### A.4    FEATURE REPRESENTATION DETAILS

As in related literature (Richter et al., 2023; Wu et al., 2024), we use amplitude-compressed and scaled complex spectrograms $\mathbf{X}_{ij} \in \mathbb{C}^{F \times T}$ as the input and output feature representations of the postfilter network with an invertible feature extractor $\Phi$:

$$\Phi(x) = \mathbf{X}_{ij} := \beta |\tilde{\mathbf{X}}_{ij}|^\alpha \exp(\mathrm{i} \cdot \angle(\tilde{\mathbf{X}}_{ij})), \quad \tilde{\mathbf{X}}_{ij} := \mathrm{STFT}(x)_{ij} \tag{24}$$

where $\angle$ denotes the phase of a complex number, and STFT is a complex-valued short-time Fourier transform (STFT). For this STFT, we use a 1534-sample (31.96 ms) Hann window resulting in $F = 768$ frequency bins and a hop length of 384 samples (74.97% overlap). We choose $\alpha = 0.3$ since we found it to produce better outputs for general audio than the original $\alpha = 0.5$ used in Wu et al. (2024), see Appendix A.7.5. Note that the choice of window length and hop length is different from ScoreDec (Wu et al., 2024) (510-sample Hann window, 320-sample hop length, 37.5% overlap) since we found the increased overlap and frequency resolution to help with output quality. Our choice of window length and hop length is the same as in the related 48 kHz speech work by Richter et al. (2024b). To keep the values of the real and imaginary parts of $\mathbf{X}$ constrained to roughly $[-1, 1]$, we set $\beta = 0.66$, which we determine as the 99.7th percentile of compressed but unscaled STFT amplitudes (i.e., Eq. (24) with $\beta = 1$) on 2,500 random clean training audio files.

### A.5    QUALITATIVE OUTPUTS FROM INITIAL DECODER

To show how the enhanced outputs by our FlowDec postfilter, $\Omega(D_0(c))$, compare to the outputs of the initial decoder $D_0(c)$ of the underlying non-adversarially trained codec NDAC-75, we show three example spectrograms in Fig. 9. The initial decoder produces overly smooth spectral structures and buzzy noise artifacts. FlowDec successfully removes these artifacts and replaces them with plausible natural spectral structures, thereby significantly enhancing the audio.

### A.6    DETAILED RESULTS FROM SUBJECTIVE LISTENING TESTS

In Fig. 10, we show the score distribution from both MUSHRA-like listening tests (Section 4.4) split by audio type. These results suggest that FlowDec may perform better on speech than DAC, particularly for FlowDec-75m versus DAC-75 at 4.5 kbit/s and that DAC may perform slightly better than FlowDec on sound files; score distributions for music are very similar.

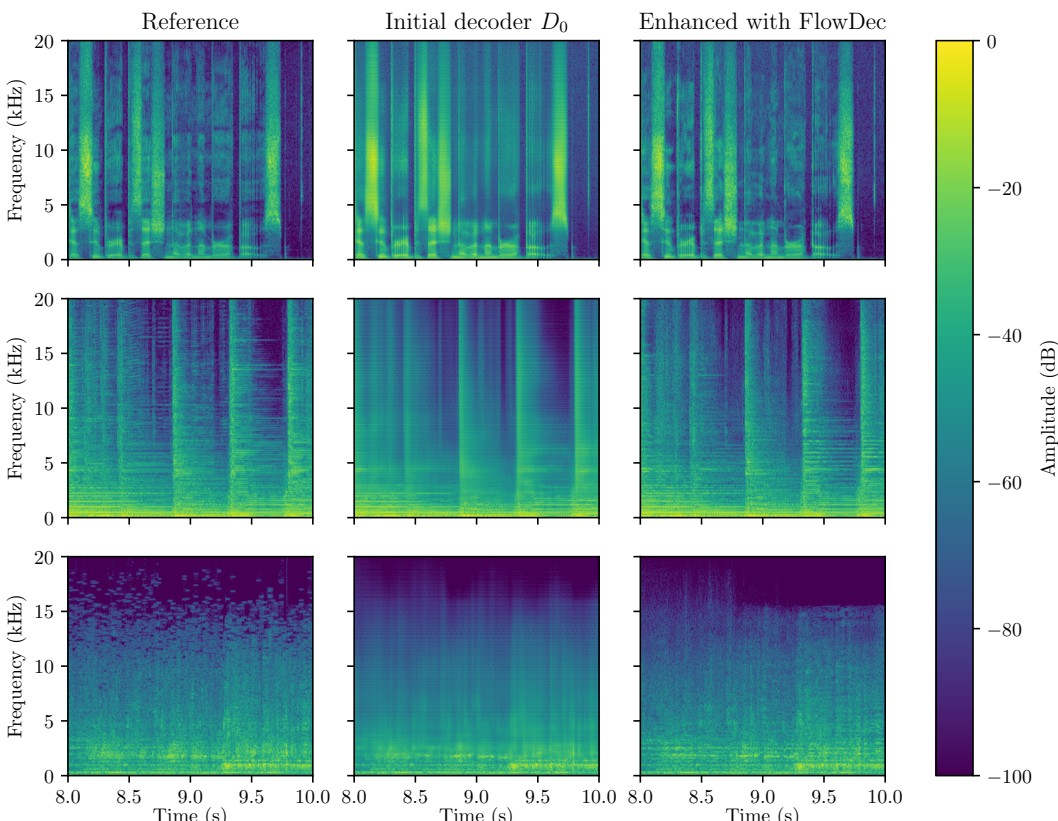

Figure 9: Spectrograms from parts of three audio examples (from top to bottom: speech, music, sound) as output by the initial decoder $D_0$ of NDAC-75, compared to their enhanced version from FlowDec-75m. The estimates from $D_0$ show severe buzzy and unnatural artifacts, which FlowDec successfully replaces with plausible spectral structures.

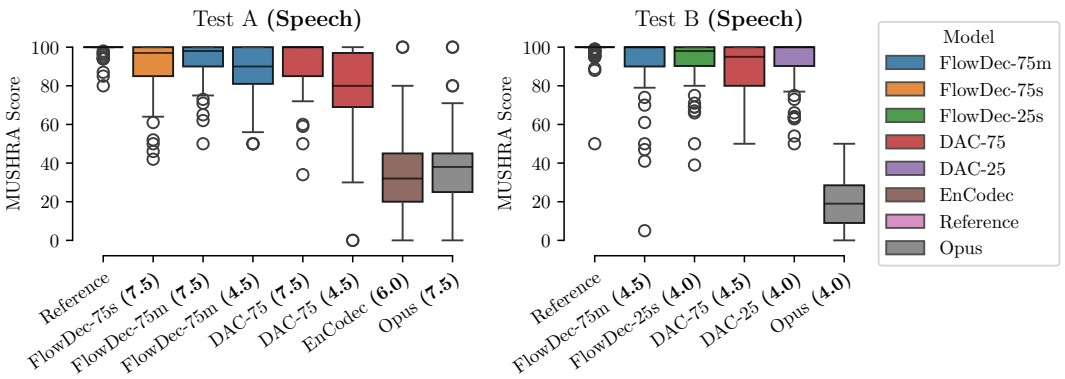

(a) **Speech samples**: Listening test results (MUSHRA score distributions)

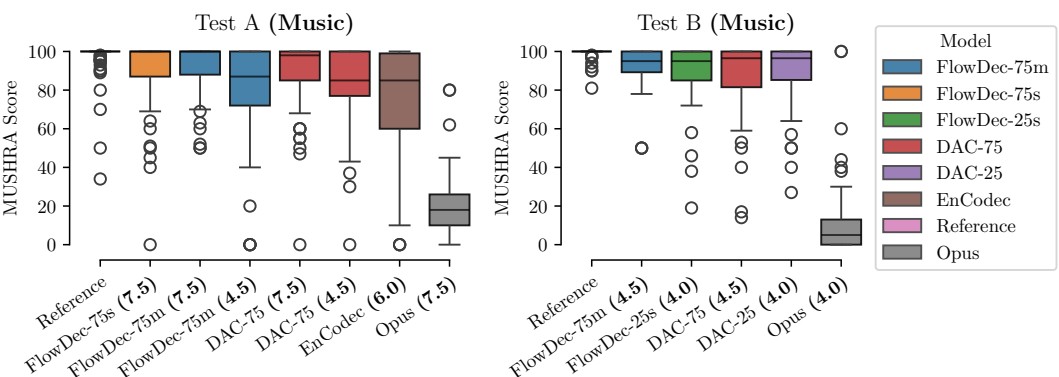

(b) **Music samples**: Listening test results (MUSHRA score distributions)

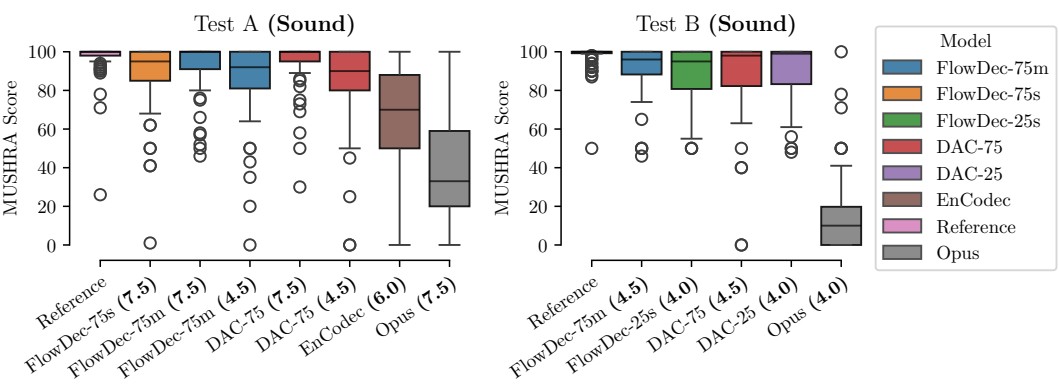

(c) **Sound samples**: Listening test results (MUSHRA score distributions)

Figure 10: Detailed results from the listening tests (Section 4.4) split by audio type.

Table 5: Mean ± 95% confidence interval of objective metrics for FlowDec(-75s) compared against baselines using the alternative ScoreDec formulation (Wu et al., 2024) or FlowAVSE (constant-$\sigma_t$) formulation, each trained on the same data with the same backbone DNN and feature representation. We show results at two different NFE (6 and 50). FAD is multiplied by 100 for readability. For "ScoreDec NC", in contrast to the original ScoreDec, we do not use the annealed Langevin corrector (Song et al., 2021) during inference, and instead double the number of predictor steps to achieve the same NFE. We can see that ScoreDec returns unusable estimates at NFE=6. At NFE=50, the metric values are now acceptable but still clearly worse than those of FlowDec in FAD, fwSSNR, SIGMOS, and logSpecMSE. In SI-SDR and SIGMOS, ScoreDec, and FlowDec achieve similar values at NFE=50.

| Method | FAD$_{\times 100}$ | SI-SDR | fwSSNR | logSpecMSE | SIGMOS |
|---|---|---|---|---|---|
| **NFE = 6** | | | | | |
| FlowDec | **1.62** | 7.55 ± 0.25 | **15.46 ± 0.07** | **80.57 ± 1.72** | **3.48 ± 0.03** |
| ScoreDec | 145.30 | -27.23 ± 0.15 | 3.15 ± 0.07 | 4873.42 ± 51.92 | 1.18 ± 0.01 |
| ScoreDec NC | 78.71 | -5.89 ± 0.19 | 4.58 ± 0.08 | 2484.17 ± 28.89 | 1.45 ± 0.01 |
| $\sigma_t = 0.05$ | 28.88 | 9.95 ± 0.21 | 5.50 ± 0.19 | 1613.40 ± 33.08 | 3.00 ± 0.02 |
| $\sigma_t = 0.66$ | 29.83 | **10.10 ± 0.22** | 6.55 ± 0.18 | 1442.94 ± 25.52 | 2.94 ± 0.02 |
| **NFE = 50** | | | | | |
| FlowDec | **1.34** | 7.41 ± 0.25 | **15.65 ± 0.06** | **81.83 ± 2.17** | 3.44 ± 0.03 |
| ScoreDec | 5.73 | 7.50 ± 0.24 | 14.45 ± 0.09 | 176.25 ± 4.12 | **3.51 ± 0.03** |
| ScoreDec NC | 3.84 | **7.56 ± 0.25** | 15.00 ± 0.08 | 130.32 ± 2.95 | 3.43 ± 0.03 |

## A.7 ABLATION STUDIES

In this appendix section, we conduct several ablation studies to further justify the choices we have made. We show comparative tables with objective metrics, and spectrograms to illustrate model behaviors qualitatively.

### A.7.1 FULL METRIC COMPARISON AGAINST SCOREDEC AND FLOWAVSE

In Table 5 we show objective metric values for FlowDec compared to the prior work ScoreDec (Wu et al., 2024) and FlowAVSE (Jung et al., 2024) at NFE=6 and NFE=50. For the baseline models here, we retrained the model with each alternative formulation while keeping all other settings (data, backbone, feature representation) the same. For FlowAVSE we train one variant with a small $\sigma_t = 0.05$, and one with the same $\sigma_t = 0.66$ as the $\sigma_y = 0.66$ setting used for FlowDec. As the metrics show, FlowDec works significantly better at NFE=6 where ScoreDec and FlowAVSE fail to produce acceptable results, and also generally outperforms ScoreDec at NFE=50.

### A.7.2 NON-ADVERSARIAL DAC WITHOUT ADDED CQT AND WAVEFORM LOSSES

As proposed in Section 3.4, we train our underlying non-adversarial codec ("NDAC") based on DAC (Kumar et al., 2024) but newly add a multiscale constant-Q transform (CQT) loss and an $L^1$ waveform-domain loss, in particular to combat the bad low-frequency preservation of the initial non-adversarial NDAC variants we trained in preliminary experiments. In Fig. 11, we show this effect qualitatively, comparing the original non-adversarial DAC without our added losses (rightmost column) to our proposed underlying codec NDAC-75 (center column), which includes these losses, in the frequency range between 0 and 1500 Hz. The original non-adversarial DAC introduces severe errors and generates a very noisy low-frequency spectrum. In comparison, our variant NDAC-75 (center column) does not suffer from these problems in the low-frequency region and produces relatively good estimates.

### A.7.3 ADVERSARIAL DAC WITHOUT AND WITH ADDED CQT AND WAVEFORM LOSSES

To further show that the advantages of our method are not caused just by the added CQT and waveform $L^1$ loss, as proposed in Section 3.4, we also train a variant of the adversarially trained DAC-75 that includes both those original adversarial losses, the original non-adversarial losses (Kumar et al.,

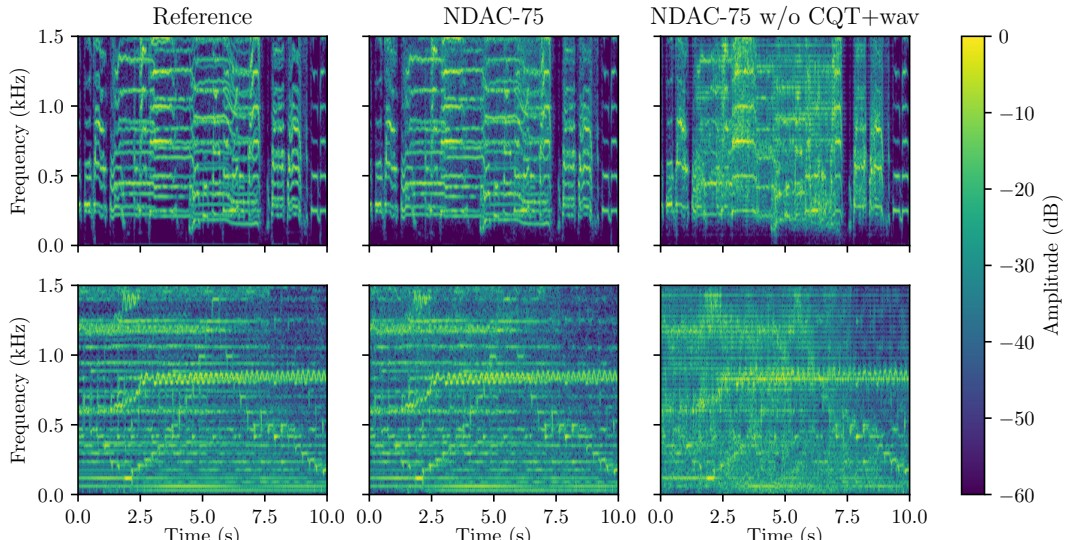

Figure 11: Low-frequency (0–1500 Hz) spectrum of two music samples (top: vocals, bottom: mixture) from our test set, comparing our underlying codec NDAC-75 against the version we initially trained without added CQT and waveform losses ("w/o CQT+wav").

Table 6: Mean ± 95% confidence interval of objective metrics for DAC-75, compared to a variant ("+Cw") trained with the original adversarial and non-adversarial losses as well as our proposed CQT and waveform losses used for DAC-75 (Section 3.4). Bitrates are in kbit/s, and FAD is multiplied by 100 for readability.

| Method | Bitrate | FAD | SI-SDR | fwSSNR | logSpecMSE | SIGMOS |
|---|---|---|---|---|---|---|
| DAC-75 | 3.00 | 9.68 | 4.66 ± 0.18 | 12.11 ± 0.07 | **80.79 ± 1.41** | **3.14 ± 0.03** |
| DAC-75 +Cw | 3.00 | **9.39** | **4.86 ± 0.19** | **12.21 ± 0.07** | 81.93 ± 1.58 | 3.09 ± 0.02 |
| DAC-75 | 4.50 | 6.80 | 6.95 ± 0.18 | 13.62 ± 0.08 | **76.95 ± 1.32** | **3.19 ± 0.02** |
| DAC-75 +Cw | 4.50 | **6.63** | **7.17 ± 0.19** | **13.74 ± 0.07** | 77.81 ± 1.49 | 3.16 ± 0.02 |
| DAC-75 | 6.00 | 5.23 | 8.54 ± 0.18 | 15.01 ± 0.08 | **74.94 ± 1.29** | **3.19 ± 0.02** |
| DAC-75 +Cw | 6.00 | **5.12** | **8.76 ± 0.19** | **15.14 ± 0.07** | 75.65 ± 1.42 | 3.17 ± 0.02 |
| DAC-75 | 7.50 | 4.15 | 10.03 ± 0.19 | 16.57 ± 0.09 | **73.05 ± 1.24** | **3.19 ± 0.02** |
| DAC-75 +Cw | 7.50 | **3.95** | **10.16 ± 0.19** | **16.65 ± 0.07** | 73.98 ± 1.38 | 3.19 ± 0.02 |

2024), and our proposed CQT and waveform $L^1$ loss. We show the metric results in Table 6. It can be seen that our proposed loss terms seem to improve FAD, SI-SDR, and fwSSNR slightly, and on the other hand, worsen logSpecMSE and SIGMOS slightly. No large differences in any metric can be seen at any particular bitrate, confirming that the strong improvements in FAD and SIGMOS of FlowDec we show in Section 5.1 are not caused purely by these loss terms being added to our underlying codec.

### A.7.4  FREQUENCY-DEPENDENT $\sigma_y$ VS. GLOBAL $\sigma_y$

In Fig. 12, we compare objective metrics of our main FlowDec-75m and FlowDec-25s variants, both with frequency-dependent $\sigma_y$, against each corresponding variant with a global $\sigma_y$ ("g$\sigma_y$"). We use each method at a bitrate of 7.5 kbit/s for FlowDec-75m and 4.0 kbit/s for FlowDec-25s, and run inference at different NFE. At a low NFE, we see that the frequency-dependent $\sigma_y$ achieves on par or better logSpecMSE and FAD scores, particularly for the 25 Hz models. For the metrics SI-SDR, fwSSNR, and SIGMOS, which choice of $\sigma_y$ is optimal seems not as clear. At NFE=4, the global $\sigma_y$ variants deteriorate significantly in logSpecMSE but gain in SI-SDR, indicating that oversmoothing

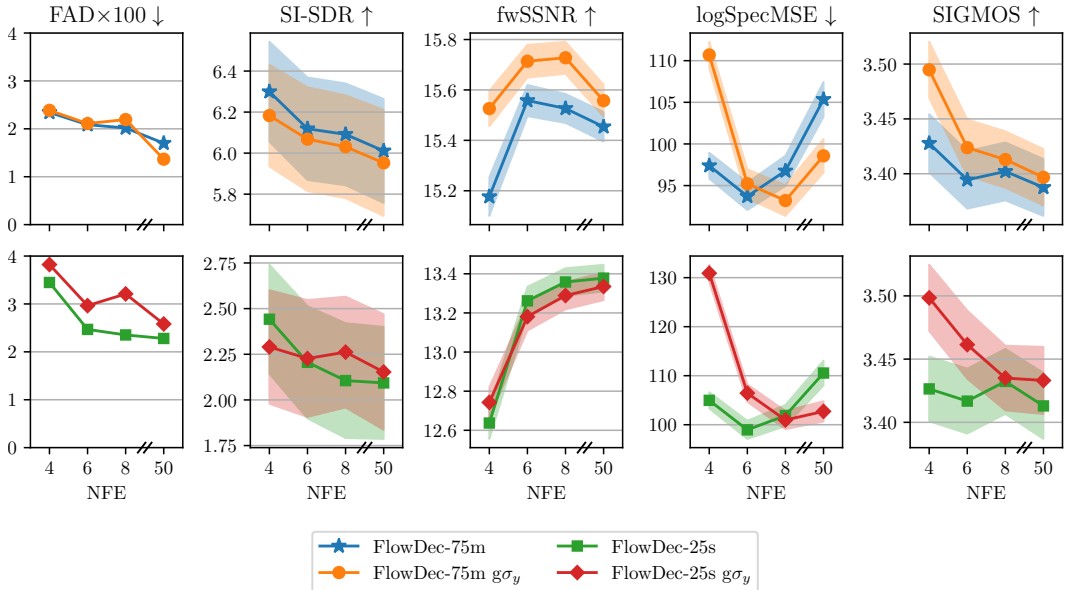

Figure 12: Objective metrics at different NFE, from our main FlowDec-75m (top row) and FlowDec-25s (bottom row) models, compared against each corresponding global $\sigma_y$ variant ("g$\sigma_y$").

Table 7: Mean ± 95% confidence interval of objective metrics for our method compared to the original NCSN++ architecture, and compared to the original choice of $\alpha = 0.5$ in contrast to our $\alpha = 0.3$. We use NFE=6 with the midpoint solver. FAD is multiplied by 100 for readability. Best in bold, second best underlined.

| Method | FAD | SI-SDR | fwSSNR | logSpecMSE | SIGMOS |
|---|---|---|---|---|---|
| FlowDec-75s | **1.62** | **7.55 ± 0.25** | **15.46 ± 0.07** | 80.57 ± 1.72 | 3.48 ± 0.03 |
| with original NCSN++ | 1.75 | 7.51 ± 0.25 | 15.30 ± 0.06 | **79.84 ± 1.76** | 3.45 ± 0.03 |
| with $\alpha = 0.5$ | 2.16 | 7.54 ± 0.25 | 14.49 ± 0.09 | 130.10 ± 1.98 | **3.57 ± 0.03** |

of high frequencies is occurring; the frequency-dependent $\sigma_y$ variants exhibit this effect much less strongly.

### A.7.5 NETWORK ARCHITECTURE AND FEATURE REPRESENTATION

In Table 7, we show metric results of FlowDec-75s, compared to two ablation model variants: one trained with the original NCSN++ architecture (Song et al., 2021; Richter et al., 2023), and one trained with the original choice of the feature representation parameter $\alpha = 0.5$ (Welker et al., 2022; Richter et al., 2023; Wu et al., 2024). We can see that FlowDec-75s performs best in FAD, SI-SDR, and fwSSNR, and significantly improves upon $\alpha = 0.5$ in logSpecMSE. In SIGMOS, which is a speech-only metric, the $\alpha = 0.5$ model achieves the best score, which may hint at $\alpha = 0.5$ being more optimal for speech signals; however, in all other metrics $\alpha = 0.3$ seems to be a better choice, and it seems to work better overall for general audio.

### A.7.6 COMPARISON OF ODE SOLVERS

In Fig. 13, we show that the numerical Midpoint ODE solver is much more effective than the simpler Euler ODE solver at producing high-quality audio at low numbers of DNN evaluations (low number of function evaluations (NFE)). Both solvers perform similarly at a high NFE of 50, but Euler generally degrades significantly at low NFE (4, 6, 8). While Euler achieves better SI-SDR, it at the same time shows significantly worse fwSSNR and logSpecMSE, which indicates spectral oversmoothing (removal of high frequencies). Midpoint performs similarly for NFE=6 as for NFE=8 and NFE=50

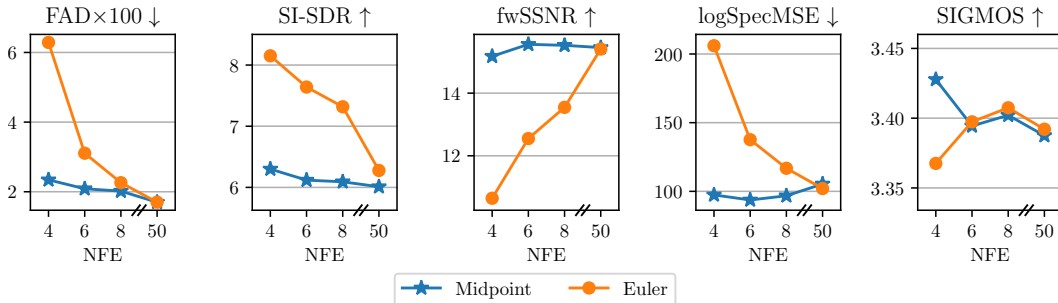

Figure 13: Objective metrics for FlowDec-75m at 7.5 kbit/s, comparing the Euler and Midpoint solver for different NFE.

but degrades slightly at the next possible lower NFE=4, thus confirming our default choice NFE=6 to be a good choice along the tradeoff between output quality and inference speed.

## A.8 QUALITATIVE SPECTROGRAM COMPARISONS

In Fig. 14, we show spectrograms comparing FlowDec-75m and DAC-75 on three examples with high harmonic content such as speech and isolated music instruments. We can see that, for these examples, FlowDec recovers more plausible natural spectral structures, and recovers high harmonics better.

For fairness, in Fig. 15, we show the three examples from our test set with the worst logSpecMSE values for FlowDec, and also those with the worst fwSSNR values in Fig. 16. We again compare FlowDec-75m against DAC-75 and also show the output from the initial decoder, NDAC-75. For the logSpecMSE examples, we see that FlowDec either inpaints frequencies that are not there in the clean reference, or wrongly removes high frequencies present in the initial decoder outputs beyond 16 kHz, which may be related to training on music data with a sampling rate of 32 kHz.

For the example with worst fwSSNR (-3.32) in Fig. 16, we can see that FlowDec mistakenly filters out most of the strong frequency content around 6 kHz even though it is present in the initial decoder output, and replaces it with spectrally more complex but wrong structures, indicating that the FlowDec postfilter is mistakenly treating these sounds as artifacts from the initial decoder rather than parts of the target signal. For the other two next-worst fwSSNR examples, FlowDec reconstructs relatively similar estimates as DAC-75, with no particularly implausible structures visible.

## A.9 FULL OBJECTIVE METRICS TABLE

In the main paper, we showed objective metrics result visually. For completeness, we list the exact numbers of metric values in Table 8.

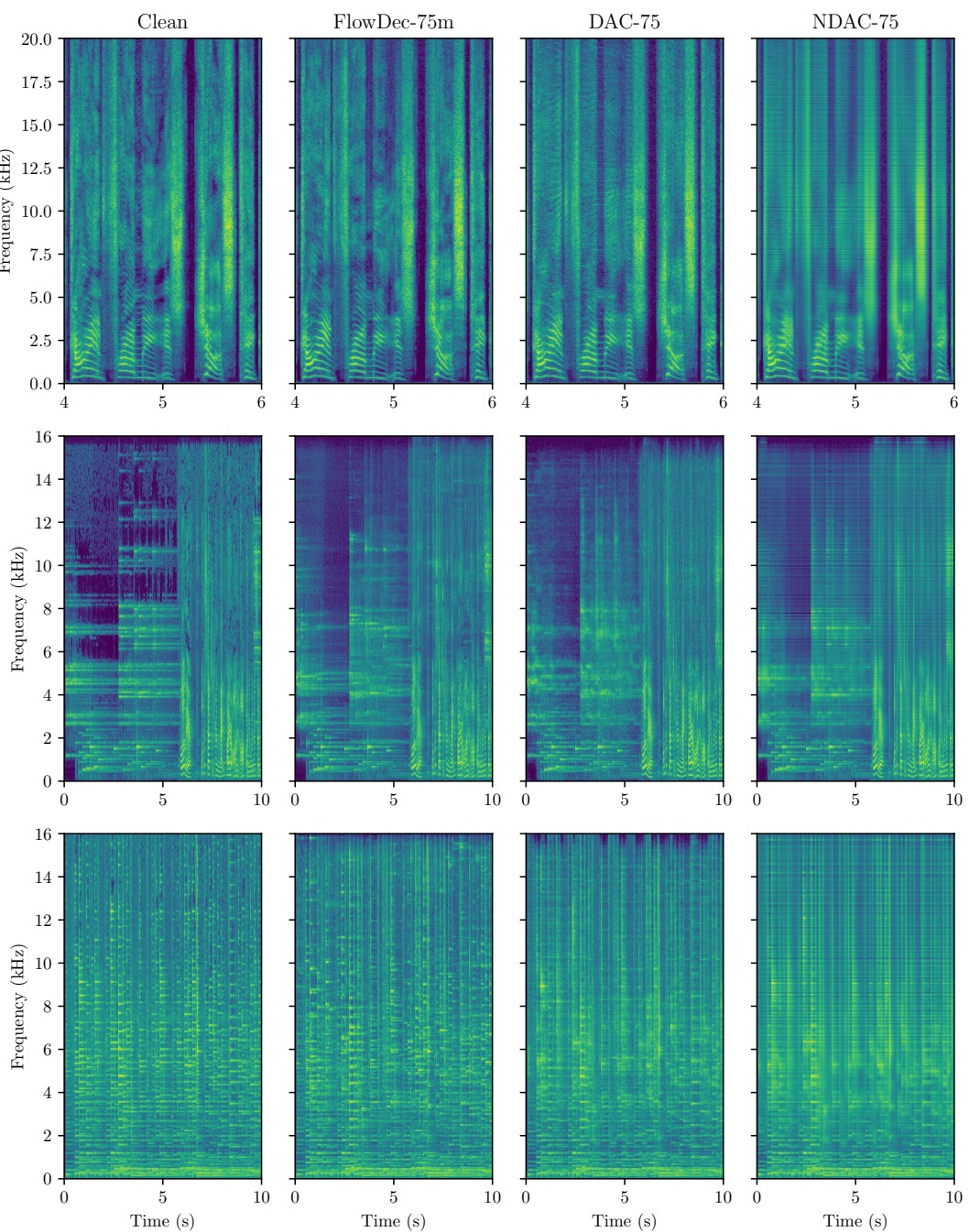

Figure 14: Spectrograms (pre-emphasis of 1.0 applied) comparing FlowDec-75m against DAC-75 and NDAC-75 on three audio files: speech (top), glockenspiel and speech (middle), and acoustic guitar (bottom).

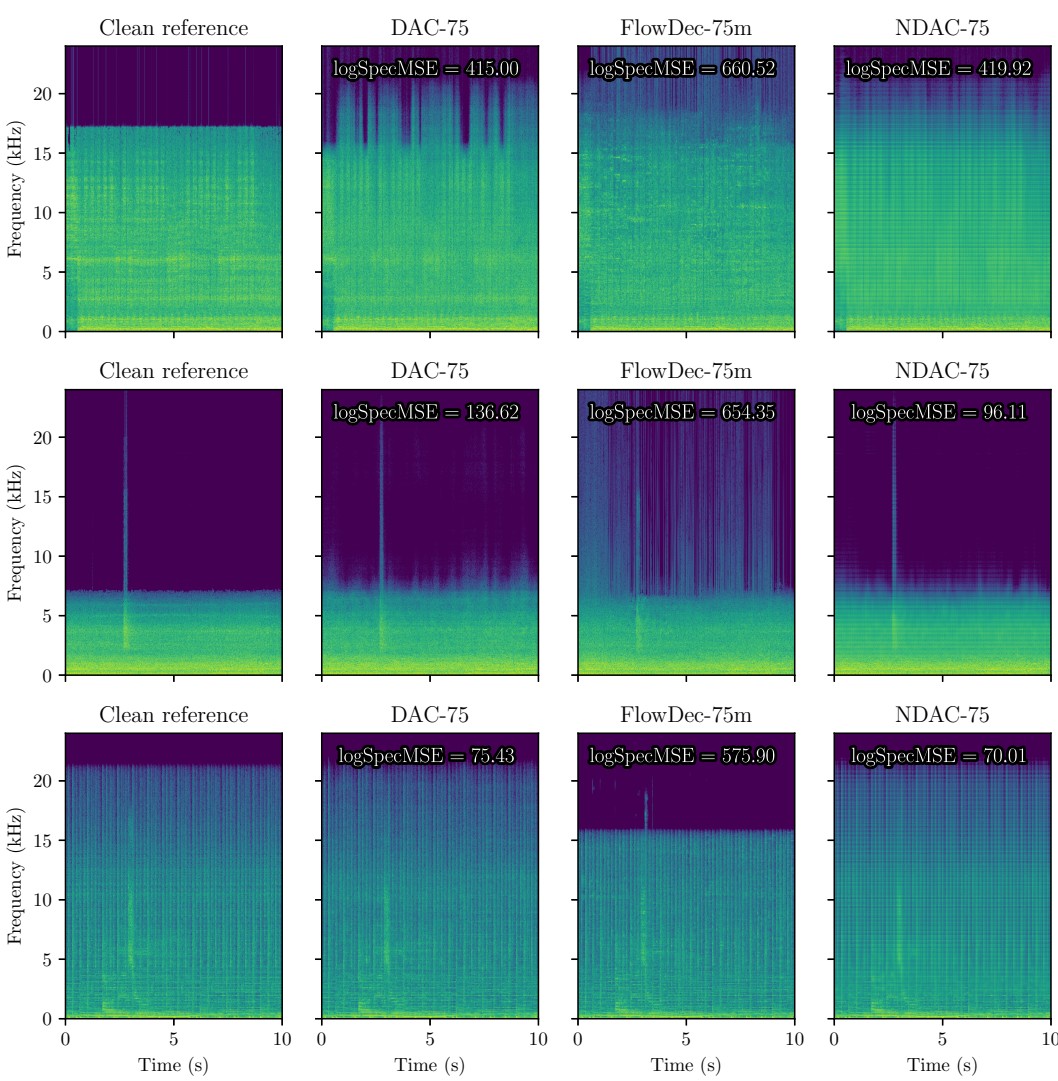

Figure 15: Spectrograms comparing FlowDec-75m against DAC-75 as well as the initial decoder output (NDAC-75) on the three audio files where FlowDec-75m produces the worst logSpecMSE values on the whole test set.

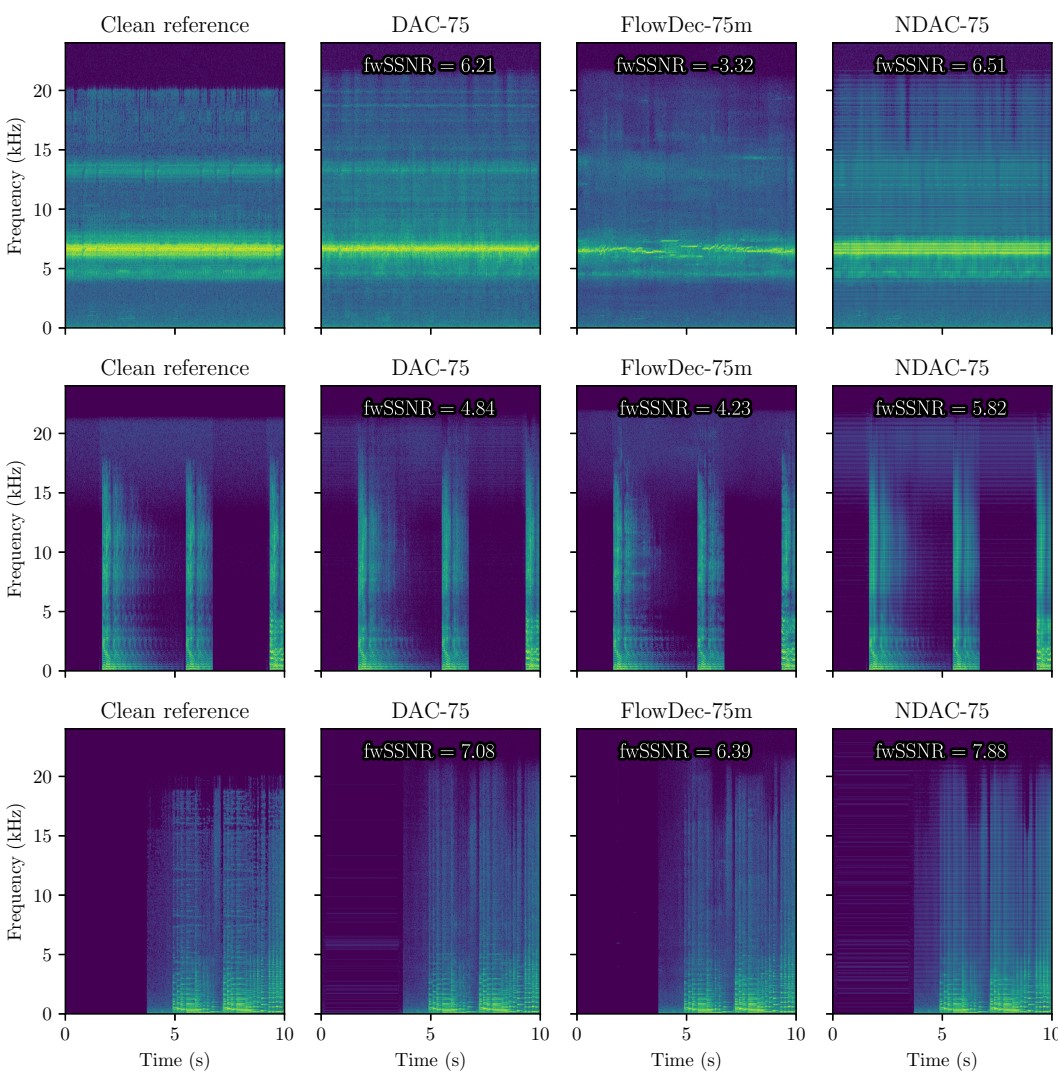

Figure 16: Spectrograms comparing FlowDec-75m against DAC-75 as well as the initial decoder output (NDAC-75) on the three audio files where FlowDec-75m produces the worst fwSSNR values on the whole test set.

Table 8: Mean ± 95% confidence interval of all metrics shown visually in Fig. 4. Best in bold, second best underlined. MBD refers to the method from San Roman et al. (2023) using their official 24 kHz checkpoint at 6 kbit/s. Bitrates in kbit/s.

| Method | Bitrate | FAD ↓ | SI-SDR ↑ | fwSSNR ↑ | logSpecMSE ↓ | SIGMOS ↑ |
|---|---|---|---|---|---|---|
| **7.50–12.00 kbit/s** | | | | | | |
| FlowDec-75m | 7.50 | 2.09 | 6.12 ± 0.25 | 15.56 ± 0.06 | 93.71 ± 1.65 | 3.39 ± 0.03 |
| FlowDec-75s | 7.50 | **1.62** | 6.22 ± 0.25 | 15.61 ± 0.07 | 93.65 ± 1.72 | **3.44 ± 0.03** |
| DAC-75 | 7.50 | 4.15 | 9.23 ± 0.19 | 16.21 ± 0.09 | 85.49 ± 1.24 | 3.16 ± 0.02 |
| 2xDAC-75 | 7.50 | 4.36 | **9.54 ± 0.19** | 17.03 ± 0.07 | 82.16 ± 1.26 | 3.19 ± 0.02 |
| DAC 44.1 kHz | 7.75 | 6.00 | 9.30 ± 0.19 | 16.46 ± 0.07 | 98.61 ± 1.83 | 3.16 ± 0.03 |
| NDAC-75 | 7.50 | 34.46 | 7.49 ± 0.26 | **17.50 ± 0.09** | **75.07 ± 1.17** | 2.71 ± 0.02 |
| EnCodec | 12.00 | 4.08 | 8.98 ± 0.18 | 13.66 ± 0.13 | 135.68 ± 2.66 | 2.61 ± 0.02 |
| **6.00–6.03 kbit/s** | | | | | | |
| FlowDec-75m | 6.00 | **2.53** | 5.16 ± 0.25 | 14.36 ± 0.06 | 95.11 ± 1.68 | **3.40 ± 0.03** |
| DAC-75 | 6.00 | 5.23 | 7.72 ± 0.18 | 14.70 ± 0.08 | 88.08 ± 1.29 | 3.16 ± 0.02 |
| 2xDAC-75 | 6.00 | 5.30 | **8.09 ± 0.19** | 15.53 ± 0.07 | 84.50 ± 1.30 | 3.18 ± 0.02 |
| DAC 44.1 kHz | 6.03 | 7.23 | 7.64 ± 0.19 | 14.76 ± 0.07 | 100.87 ± 1.84 | 3.16 ± 0.03 |
| NDAC-75 | 6.00 | 36.80 | 6.36 ± 0.24 | **15.74 ± 0.08** | **76.54 ± 1.21** | 2.70 ± 0.02 |
| EnCodec | 6.00 | 7.35 | 6.27 ± 0.18 | 11.74 ± 0.12 | 142.69 ± 2.71 | 2.41 ± 0.02 |
| MBD (24 kHz) | 6.00 | 16.87 | -0.75 ± 0.42 | 7.97 ± 0.10 | 782.25 ± 15.9 | 2.73 ± 0.02 |
| **4.31–4.50 kbit/s** | | | | | | |
| FlowDec-75m | 4.50 | **3.01** | 3.57 ± 0.24 | 12.95 ± 0.07 | 99.02 ± 1.76 | **3.41 ± 0.03** |
| DAC-75 | 4.50 | 6.80 | 6.08 ± 0.18 | 13.33 ± 0.08 | 90.19 ± 1.32 | 3.15 ± 0.02 |
| 2xDAC-75 | 4.50 | 6.42 | **6.47 ± 0.18** | 14.20 ± 0.07 | 87.44 ± 1.36 | 3.16 ± 0.02 |
| DAC 44.1 kHz | 4.31 | 9.25 | 5.74 ± 0.19 | 13.23 ± 0.08 | 104.08 ± 1.85 | 3.13 ± 0.03 |
| NDAC-75 | 4.50 | 41.01 | 5.05 ± 0.24 | **14.41 ± 0.08** | **78.36 ± 1.23** | 2.68 ± 0.02 |
| **2.58–3.00 kbit/s** | | | | | | |
| FlowDec-75m | 3.00 | **4.41** | 1.10 ± 0.28 | 11.43 ± 0.06 | 104.62 ± 1.87 | **3.40 ± 0.03** |
| DAC-75 | 3.00 | 9.68 | 3.83 ± 0.18 | 11.83 ± 0.07 | 94.81 ± 1.41 | 3.10 ± 0.03 |
| 2xDAC-75 | 3.00 | 8.82 | **4.29 ± 0.19** | 12.74 ± 0.07 | 91.79 ± 1.43 | 3.13 ± 0.03 |
| DAC 44.1 kHz | 2.58 | 12.82 | 2.78 ± 0.20 | 11.24 ± 0.08 | 110.00 ± 1.88 | 2.93 ± 0.03 |
| NDAC-75 | 3.00 | 49.07 | 2.64 ± 0.26 | **12.89 ± 0.08** | **81.75 ± 1.28** | 2.59 ± 0.02 |
| EnCodec | 3.00 | 15.66 | 3.44 ± 0.20 | 9.74 ± 0.12 | 153.48 ± 2.78 | 2.10 ± 0.02 |
| **4.00 kbit/s (25 Hz)** | | | | | | |
| FlowDec-25s | 4.00 | **2.47** | 2.21 ± 0.31 | 13.26 ± 0.07 | 98.95 ± 1.85 | **3.42 ± 0.03** |
| DAC-25 | 4.00 | 5.98 | 6.15 ± 0.21 | 13.57 ± 0.08 | **89.90 ± 1.34** | 3.13 ± 0.03 |
| 2xDAC-25 | 4.00 | 5.96 | **6.49 ± 0.21** | **13.95 ± 0.08** | 90.45 ± 1.27 | 3.17 ± 0.02 |

