# OpenReview forum: "FlowDec: A flow-based full-band general audio codec with high perceptual quality"
_ICLR.cc/2025/Conference — ICLR 2025 Poster_

### Official Review · Reviewer_bRLZ · 2024-10-20

**Soundness:** 2
**Presentation:** 2
**Contribution:** 2
**Rating:** 6
**Confidence:** 4

**Summary:**

The authors introduce FlowDEC, a flow matching-based post-filtering method designed to enhance the audio quality decoded from discrete audio tokens. This method has demonstrated strong objective and subjective results.

**Strengths:**

- Possibly the first application of flow matching in postfiltering.
- Beautiful experiment result plots and strong objective and subjective results.
- Enhance DAC by incorporating a multi-scale Constant-Q Transform (CQT) loss.

**Weaknesses:**

- The latency of the proposed method is at least an order of magnitude higher than that of DAC. To facilitate a comprehensive evaluation, it would be beneficial to receive specific latency measurements for both the proposed method and the DAC. Furthermore,  identifying the applications or use cases that could be critically impacted by this higher latency would provide valuable context. Moreover, to support the validation of the method’s performance, it would be advantageous if the authors could provide an online demo that allows for direct comparison of the latency between the two methods.
- The baselines are limited. There are other diffusion-based works that reconstruct audio waveforms from discrete tokens, e.g. Multi-band diffusion [1].
- The concept of selecting a data-dependent prior, as proposed, is not entirely novel, with works such as Priorgrad [2] having implemented a similar strategy. To better understand the relationship between the proposed method and existing approaches, it would be helpful to have a clearer comparison of their similarities and differences.
- Multi-band diffusion [1] uses frequency-dependent noise levels, which is a notable feature that could be further discussed in comparison to the proposed method.

[1] "From Discrete Tokens to High-Fidelity Audio Using Multi-Band Diffusion" by Robin San Roman, Yossi Adi, Antoine Deleforge, Romain Serizel, Gabriel Synnaeve, and Alexandre Défossez.

[2] "PriorGrad: Improving Conditional Denoising Diffusion Models with Data-Dependent Adaptive Prior" by Sang-gil Lee, Heeseung Kim, Chaehun Shin, Xu Tan, Chang Liu, Qi Meng, Tao Qin, Wei Chen, Sungroh Yoon, Tie-Yan Liu

**Questions:**

-  The question arises as to why the waveform is not reconstructed directly from discrete tokens using flow matching. To provide a more comprehensive evaluation, could you elaborate on the potential computational or quality trade-offs between direct reconstruction via flow matching and the method proposed in the paper?
- Regarding line 168-171, the presence of $p_{data}(\cdot|y)$ is not clear as it seemingly should just be $p_{data}(x)$. This needs further clarification and justification.
- From line 220 to 223 and in Figure 2, it may be inappropriate as we cannot obtain $x_1$ for real postfiltering problems. This aspect requires reconsideration or better explanation.
- In Figure 3, the use of a large $\sigma_t$ of 0.1 o illustrate that FlowAVSE is non-contractive is not convincing, especially since the original FlowAVSE employs a value of 0.04. It would be beneficial for the authors to provide a clearer explanation for this choice or consider revising the value used.
- As $y$ is a time-domain signal, it is unclear how to apply frequency-dependent $\sigma_y$ since it cannot be directly applied. The authors need to provide more details on this aspect.

---

> ### Author Response · Authors · 2024-11-21
>
> We thank the reviewer for their positive comments on our results and illustrations and their note that we are the first work to apply flow matching for postfiltering. Regarding the reviewer's mentioned weaknesses, see below:
>
> > The latency of the proposed method is at least an order of magnitude higher than that of DAC. [...] it would be advantageous if the authors could provide an online demo that allows for direct comparison of the latency. [...] identifying the applications or use cases that could be critically impacted by this higher latency would provide valuable context.
>
> **Please note that neither DAC nor FlowDec are targeted towards real-time scenarios and do not currently offer streaming variants.** The DNN architectures of both methods involve non-causal operations. This also prohibits the preparation of a meaningful online demonstration of each model’s latency. We provide measurements of the real-time factor (RTF) in section 5.3 in our work, showing that FlowDec (RTF = 0.23) significantly improves over ScoreDec (RTF = 1.70), paving the way for future streamable developments.
>
> **We now more explicitly note the lack of streamability in our conclusion and mention applications unlocked by future streamable modifications.** Note that it is possible to define a diffusion- and flow-based postfilter for any streamable codec, as we note in our conclusion. Our focus with FlowDec is on the attainable perceptual quality of modern neural codecs when removing adversarial training and using a flow matching postfilter instead. Hence, we leave streamable variants for future work.
>
> >The baselines are limited. There are other diffusion-based works that reconstruct audio waveforms from discrete tokens, e.g. Multi-band diffusion [1].
>
> Thank you for this reference. In the demo page we now provide [here](https://flowdec2024.github.io/FlowDecSupplementary/), we show results for MBD [1] using the published checkpoints (https://github.com/facebookresearch/audiocraft/blob/main/docs/MBD.md) at 6.0 kbit/s and 24 kHz. We follow the authors’ recommendation [discussed here](https://github.com/facebookresearch/audiocraft/issues/236) to avoid residual noise. Note that the audio quality is noticeably worse than DAC and FlowDec. The metric results from MBD on our test set are:
>
> Metric | Value
> --- | ---
> FAD x100 	| 16.87
> SI-SDR	| -0.75 +/- 0.42
> fwSSNR	| 7.97 +/- 0.10
> logSpecMSE	| 782.25 +/- 15.91
> SIGMOS	| 2.73 +/- 0.02
>
> **These results indicate that MBD performs both qualitatively and quantitatively worse than FlowDec and DAC.** We added these metrics to Table 8 in the Appendix, but do not include these results in the main paper as there is no full-band model available for MBD, so a direct comparison is not fair. Note that MBD is slower than FlowDec with an RTF of 0.71 compared to our RTF of 0.23, see their table A.7.
>
> >The concept of selecting a data-dependent prior, as proposed, is not entirely novel, with works such as Priorgrad [2] having implemented a similar strategy. [...]
>
> The reviewer is correct that PriorGrad also makes use of a data-dependent prior, as do the prior works SGMSE and ScoreDec we build upon and cite (see section 2.2). **We now state this more explicitly in section 2.2 and add a citation for PriorGrad.** A more in-depth comparison:
>
> * ScoreDec uses the SGMSE formulation, which uses score matching with an interpolating mean $\mu_t$, but suffers from prior mismatch (i.e., the prior is different during training and inference, see lines 244-249), exhibits a more highly curved flow field (Figure 3), performs worse than FlowDec empirically, and has more costly inference (Table 4).
> * PriorGrad uses an ELBO loss based on Denoising Diffusion Probabilistic Models (DDPM). This formulation also has prior mismatch: the forward process does not reach mean 0 since $\sqrt{\bar{\alpha}_t} \neq 0$ at $t=T$, but $z$ is nonetheless sampled from a zero-mean Gaussian during inference. Further, PriorGrad defines the diffusion in the space of the residual $x_0 - \mu$. Revisiting such a residual-based formulation, perhaps also using flow matching, would be an interesting future work.
>
> > Multi-band diffusion [1] uses frequency-dependent noise levels, which is a notable feature that could be further discussed in comparison to the proposed method.
>
> We thank the reviewer for this observation. MBD indeed also uses band-dependent noise levels. **We now accordingly mention this fact and add a reference to MBD in section 3.5.** A key difference is that MBD uses different noise levels on only 4 broad Mel bands, while FlowDec uses a highly granular $\sigma_y(f)$ for each of 786 STFT bands. Further, MBD determines the noise level based on the per-band variance of the clean data, while we determine the noise level based on the per-band errors of the initial decoder. This can be interpreted to serve a dual purpose: matching the natural power distribution of audio signals, and adapting the noising process to the initial decoder's errors.

---

> ### Author Response · Authors · 2024-11-21
>
> Addressing each of the reviewer's questions:
>
> >The question arises as to why the waveform is not reconstructed directly from discrete tokens using flow matching. To provide a more comprehensive evaluation, could you elaborate on the potential computational or quality trade-offs between direct reconstruction via flow matching and the method proposed in the paper?
>
> We thank the reviewer for this good question. By reconstructing “directly from discrete tokens”, we assume that the reviewer means to use a flow model that generates audio data by flowing from a simple prior such as a zero-mean standard Gaussian and is conditioned on discrete tokens from a neural codec, as done in MBD. Note that our flow model is also conditioned on codec tokens since we pass the decoder prediction $Y = D_0(c)$ as conditioning (lines 262-264).
>
> **A key difference from the approach followed in MBD is the mean-shifted data-dependent prior we employ.** To be able to use such a data-dependent prior, we aim to use a decent initial estimate in a mean sense, and providing this estimate is the main purpose of our initial decoder $D_0$.
>
> **The advantages of our approach are:**
> * **Our data-dependent prior enables the inference speed advantages** discussed in section 3.2. Note, for example, that MBD uses 20 steps for inference whereas we find that 6 steps are sufficient for FlowDec.
> * **The mapping from an initial estimate to the target signal is much simpler than the mapping from pure noise**, which should lead to more stable training. This should also result in a flow field that is less tangled and less curved (see Fig. 2 and Fig. 3 in our paper, and Fig. 1 in Tong et al., 2024).
> * **It is less likely that the model inadvertently ignores the conditioning** since the inference process itself starts from an estimate close to the ground truth, rather than from pure noise; this may also avoid hallucinations.
>
> **Possible downsides are:**
> * **When the initial decoder provides a bad estimate, the postfilter may fail to properly enhance the audio.** We however did not observe such an effect in our experiments.
> * **Training a two-stage model may take longer overall** (longer overall training duration for initial decoder + postfilter). To test this would require a direct comparison of otherwise equal one-stage and two-stage models.
> * **A two-stage model may be slower for inference since two DNNs are used.** In our experience however, the runtime is dominated by the flow model, not by the initial decoder (see section 5.3), so using only one flow-based stage would not significantly improve inference speed.
>
> >Regarding line 168-171, the presence of pdata(.|y) is not clear as it seemingly should just be pdata(.). This needs further clarification and justification.
>
> We thank the reviewer for their comment. **$p_{\text{data}}(\cdot|y)$ is the correct notation here**, since the model $\Omega$ is a postfilter. $\Omega$ should thus not produce unconditional samples $x \sim p_{\text{data}}(x)$ as in unconditional image or audio generation tasks, but should rather produce conditional samples from the distribution of clean audios given a particular code $c$, i.e., $x \sim p_{\text{data}}(x|c)$.
> This is similar to, for instance, class- or text-conditional image diffusion models producing samples from $x \sim p_{\text{data}}(x|C)$ where $C$ is a text embedding or image class. We can equivalently write $p_{\text{data}}(x|y) = p_{\text{data}}(x|c)$ since $y = D_0(c)$ and $D_0$ is known, deterministic and non-compressive, see lines 172-174. We choose the notation $p_{\text{data}}(x|y)$ to make the dependence on the estimate from the initial decoder, $y = D_0(c)$, explicit.
>
> >From line 220 to 223 and in Figure 2, it may be inappropriate as we cannot obtain for real postfiltering problems. This aspect requires reconsideration or better explanation.
>
> **The purpose of this paragraph is to make an explicit but intuitive link to (Pooladian et al., 2023 and Tong et al., 2024) regarding minibatch optimal transport.** Here we show intuitively that our data-dependent joint flow formulation avoids overlapping priors and tangled flow fields between different $x_1$ (compare with Figure 1 in Tong et al., 2024) by construction while avoiding the need to perform minibatch optimal transport calculations during training. The reviewer is correct that the actual marginal distribution cannot be used in practice, and we have noted in the manuscript that “we do not use it for inference or training”.

---

> ### Author Response · Authors · 2024-11-21
>
> >In Figure 3, the use of a large sigma_y of 0.1 to illustrate that FlowAVSE is non-contractive is not convincing, especially since the original FlowAVSE employs a value of 0.04. It would be beneficial for the authors to provide a clearer explanation for this choice or consider revising the value used.
>
> We understand your concern. **Please note that the specific value of $\sigma_t$ for FlowAVSE has no bearing on the flow field being non-contractive or not.** The FlowAVSE flow field is non-contractive due to the independence of the target flow from the sampled noise since it uses I-CFM from Tong et al., 2024 (Eqns. 14 and 15), and this holds for any choice of $\sigma_t$. **We have added a note in our section 3.2 to clarify this.**
>
> Since Fig. 3 shows a toy problem, we chose $\sigma_t = 0.1$ for visual reasons since the illustration is less visually clear for the smaller $\sigma_t = 0.04$, see the figure with a smaller $\sigma_t$ which we provide at [this GitHub link](https://github.com/FlowDec2024/FlowDecSupplementary/blob/09dc2fa8911e08feaffedfae0eed155460f35b70/fig3-smaller-stdev-for-flowavse.pdf).
>
>
> >As is a time-domain signal, it is unclear how to apply frequency-dependent since it cannot be directly applied. The authors need to provide more details on this aspect.
>
> **Please see lines 259-269: We formulate the diffusion process and flow model in the complex STFT domain in practice.** This allows for a straightforward implementation of the frequency-dependent noise. For any complex spectrogram $X$ of shape $(T, F)$ where $T$ denotes time and $F$ denotes frequency, we scale each band with frequency index $f = 0, \ldots, F-1$ in the sampled noise $Z \in \mathbb{C}^{T \times F}$ by multiplying with $\sigma_t(f)$, which depends on $f$ and is constant for all time indices $t$. We will provide our code that includes the exact implementation for this frequency-dependent $\sigma_y$.

---

> > ### Comment · Reviewer_bRLZ · 2024-11-27
> >
> > Thank you for your detailed response. The comparison with MBD clearly illustrates the differences.

---

### Official Review · Reviewer_QBQt · 2024-10-30

**Soundness:** 3
**Presentation:** 4
**Contribution:** 3
**Rating:** 8
**Confidence:** 5

**Summary:**

FlowDec is an improved version of ScoreDec (uses a score-based generative model as a postfilter), by switching the objective to flow matching. It further proposes a joint flow matching objective tailored for the postfiltering task (e.g. mean-shifted noise with frequency-dependent diagonal covariance). This makes it faster than real time (unlike ScoreDec) as a practically viable, full-band, and universal audio codec without adversarial training.

**Strengths:**

* Contrary to adversarial training (a dominant approach in vocoder/codec) which requires domain specific expertise to stabilize training and tune the hyperparameters of multiple losses, FlowDec simplifies the trianing pipeline by elimiating the adversarial losses. In my opinion, this can be considered as one of the first work that achieves competitive quality to GAN-based models with RTF < 1.

* The proposed design choice is well justified overall, both in the theoretical and empericial perspective. It provides enough rigor in experimental detail, including toy experiments and detailed ablation study of each component.

**Weaknesses:**

* While the evaluation results from the paper look convincing overall, no demo page has been provided at submission. It's hard to form the reviewer's opinion on the subjective quality without access to the demo. I hope the authors can provide the samples for the reader to evaluate the subjective quality themselves.

* Although it improved the RTF by a large margin, it still requires multiple NFE from the postfilter resulting in RTF of 0.22-0.23 which is considerably slower than recent, feed-forward audio codecs.

**Questions:**

There have been several concurrent works adopting flow matching with similar task (i.e. mel spectrogram vocoder and discrete codec decoder), such as RFWave [1] and PeriodWave [2,3]. These work directly utilize flow matching as the decoder rather than postfilter, with several configs being order of magnitude faster than this work.

While the direct comparison may not be strictly necessary, can the authors provide conceptual comparisons between the aforementioned methods along with possible advantages of FlowDec?

[1] Liu, Peng, and Dongyang Dai. "RFWave: Multi-band Rectified Flow for Audio Waveform Reconstruction." arXiv preprint arXiv:2403.05010 (2024).

[2] Lee, Sang-Hoon, Ha-Yeong Choi, and Seong-Whan Lee. "PeriodWave: Multi-Period Flow Matching for High-Fidelity Waveform Generation." arXiv preprint arXiv:2408.07547 (2024).

[3] Lee, Sang-Hoon, Ha-Yeong Choi, and Seong-Whan Lee. "Accelerating High-Fidelity Waveform Generation via Adversarial Flow Matching Optimization." arXiv preprint arXiv:2408.08019 (2024).

---

> ### Author Response · Authors · 2024-11-21
>
> We would like to thank the reviewer for their positive evaluation of our work. We address the comments in detail below:
>
> >While the evaluation results from the paper look convincing overall, no demo page has been provided at submission. It's hard to form the reviewer's opinion on the subjective quality without access to the demo. I hope the authors can provide the samples for the reader to evaluate the subjective quality themselves.
>
> **We now provide a demo page at https://flowdec2024.github.io/FlowDecSupplementary/**.
>
> >There have been several concurrent works adopting flow matching with similar task (i.e. mel spectrogram vocoder and discrete codec decoder), such as RFWave and PeriodWave [...] While the direct comparison may not be strictly necessary, can the authors provide conceptual comparisons between the aforementioned methods along with possible advantages of FlowDec?
>
> Thank you for providing references for these interesting concurrent works. We are happy to discuss some conceptual and empirical similarities and differences here:
>
> * **Regarding inference speed, we have the lowest NFE of the distillation-free methods (RFWave: 10, PeriodWave: 16, FlowDec: 6).** Please note that we report RTF which is the reciprocal of the “synthesis speed” reported in PeriodWave. Here we perform similarly to PeriodWave (synthesis speed of 7.48x for PeriodWave, or 5.12x for PeriodWave-MB) as our reported RTF value of 0.2285 means a synthesis speed of 4.38x. PeriodWave-Turbo further improves the synthesis speed through adversarial distillation, whereas RFWave greatly improves it through band-parallel processing. One can conjecture that the adversarial distillation may lead to training instabilities or output artifacts similar to adversarial training, but this remains to be seen in a direct comparison. A direct NFE/speed comparison is not straightforward since PeriodWave and PeriodWave-Turbo only report results for vocoder tasks, not reconstruction from codec tokens, and work with 24 kHz audio whereas we work with 48 kHz. Further, these works evaluate different network architectures on different GPUs. In any case, **both techniques for improving inference speed (distillation, band-parallel processing) could also be applied to FlowDec.**
>
> * **RFWave uses an unconditional prior, which may potentially slow down inference** due to more curved flow fields and thus higher required NFE. **PeriodWave and PeriodWave-Turbo use an energy-based data-dependent prior inspired by PriorGrad**, specifically a zero-mean Gaussian with a coarsely data-dependent variance $\Sigma$ based on the frequency-averaged Mel spectrogram (see their section 3.3). **FlowDec instead shifts the prior’s mean to the initial prediction $y$, which results in the theoretical advantage of non-overlapping priors** we illustrate in Fig. 2.
>
> * The PeriodWave authors make interesting observations about limited high-frequency modeling (see their section 3.4) and propose several techniques to address this. Their ideas on using a Discrete Wavelet Transform and a FreeU architecture could be combined with a future version of FlowDec. Interestingly however, we did not observe any significant problems modeling high-frequency content when using the midpoint solver and NFE >= 8 even without a frequency-dependent $\sigma_y$, see our Fig. 12.

---

> > ### Comment · Reviewer_QBQt · 2024-11-26
> >
> > Thank you for providing the demo page and the conceptual comparison to concurrent work.
> >
> > * The perceptual quality is competitive with leading GAN-based methods (DAC) overall In my subjective opinion, especially under low frame & bitrate. At higher frame & bitrate I slightly prefer the sound signature of DAC due to its crispness and FlowDec having occasional warbling artifacts, but the difference seems marginal.
> >
> > * Thank you for articulating the conceptual comparison; distillation in PeriodWave, band-parallel processing in RFWave (claimed ~153x inference speed), and data-dependent prior in PriorGrad. I agree these techniques are orthogonal and can also be explored based on FlowDec as well. Improving the inference speed close to leading GAN-based methods (e.g., 30 - 50x) will likely make flow-based codec significantly more practical.
> >
> > * FlowDec's ability for high-frequency modeling is interesting; to the best of my knowledge, practitioners have commonly encountered challenges in high-frequency reconstruction when using diffusion or flow-based method on audio waveform. It would be interesting to explore whether a choice of solver (sampler) brings a significant impact on high-frequency modeling.
> >
> > Overall, I appreciate and acknowledge the author's response.

---

### Official Review · Reviewer_VB3E · 2024-11-01

**Soundness:** 3
**Presentation:** 4
**Contribution:** 3
**Rating:** 6
**Confidence:** 3

**Summary:**

The paper introduces FlowDec, a neural audio codec with a two-stage approach: (1) an autoencoder with residual vector quantization, trained without adversarial loss, and (2) a postfilter that reduces coding artifacts and enhances perceptual quality. FlowDec adapts conditional flow matching for signal enhancement, achieving improvements over previous score- and flow-based models. Both listening tests and objective metrics show that FlowDec provides perceptual quality competitive to state-of-the-art GAN-based codecs.

**Strengths:**

- The paper is well-written, with clear explanations and informative illustrations that help clarify key points.
- It provides useful context relative to previous score-based models. It also includes valuable theoretical insights and formalism to support its approach.
- It performs on par with state-of-the-art GAN codecs and shows slight improvements for speech signals. FlowDec may handle high-frequency harmonics better, where GAN-based codecs often introduce periodicity and harmonic artifacts.

**Weaknesses:**

The paper introduces FlowDec as a codec, but its primary focus seems to be a flow-based postfilter that enhances outputs from an *underlying* codec. The main weakness of this paper is the decision not to explore a GAN-based codec combined with the proposed postfilter. The paper argues against adversarial training in Section 3.3, stating that the “generated phase may be very different from the original phase.” However, FlowDec itself doesn't seem to preserve phase. Results presented in Section 5.1 show that FlowDec performs worse than GAN-based codecs in terms of SI-SDR and fwSSNR, which are sensitive to phase shifts. This might still be acceptable for a low-bitrate codec, particularly if perceptual quality is prioritized. However, the paper should more clearly justify why a GAN isn't used as the underlying codec. A GAN-based codec with a flow-based postfilter might not only reduce coding artifacts but also achieve higher reconstruction metrics. This would call into question the rationale behind the proposed non-adversarial DAC (NDAC) model. Overall, the distinction between FlowDec as a novel audio codec and its role as an enhancement tool for *any* audio codec could be clarified.

**Questions:**

1. Could the authors clarify the main purpose of FlowDec? Why is it presented as a standalone codec rather than positioning the postfilter as an enhancement tool?
2. What is the rationale for choosing to enhance only outputs from a non-adversarial codec instead of using GAN output as the initial estimate for FlowDec's postfilter? Could the authors specify what benefits NDAC offers over DAC in this proposed two-stage setup?
3. The STFT settings seem unconventional. A 1534-point FFT would definitely be slower than a 1536-point FFT due to FFT efficiency with highly composite sizes. I assume the goal is to get exactly 768 frequency bins, but why not just use a 1536-point FFT with 769 bins? Overall, this may be negligible given the slower inference, but I'm curious about the tradeoff here.

---

> ### Author Response · Authors · 2024-11-21
>
> We thank their reviewer for their in-depth evaluation of our work. Please find our detailed response below.
>
> >The main weakness of this paper is the decision not to explore a GAN-based codec combined with the proposed postfilter. The paper argues against adversarial training in Section 3.3, stating that the “generated phase may be very different from the original phase.” However, FlowDec itself doesn't seem to preserve phase. [...]
>
> We thank the reviewer for their valuable comment on phase modeling. While ScoreDec (Wu et al., 2024) found improved phase compared to AudioDec, after investigating further, we found that **there is indeed no clear advantage regarding phase modeling of either FlowDec or DAC when compared to each other**, and DAC already achieves very good phase reconstruction. This may be due to DAC’s quantization factorization technique, achieving especially high codebook usage. **We have thus revised our motivation in section 3.3 to focus more on other issues of adversarial training.** As far as we can tell, FlowDec's lower SI-SDR and fwSSNR are mainly caused by increased high-frequency generation, which is penalized by intrusive metrics even when perceptually convincing.
>
> >Could the authors clarify the main purpose of FlowDec? Why is it presented as a standalone codec rather than positioning the postfilter as an enhancement tool?
>
> **The main purpose of FlowDec is to show that postfiltering with a generative flow matching postfilter is a competitive alternative to SOTA GAN-based approaches, which is a novel result to the best of our knowledge.**
>
> **The flow-based postfiltering ideas we present in this work can indeed be applied to any underlying codec**. What we name “FlowDec” is our overall proposed and evaluated system which, as a combination of a non-adversarial initial decoder (NDAC) and a postfilter, functions as an overall codec. “FlowDec” is hence a codec building upon DAC in the same sense that DAC is a codec building upon RVQGAN [1*]: by removing, replacing, and adding components.
>
> > What is the rationale for choosing to enhance only outputs from a non-adversarial codec instead of using GAN output as the initial estimate for FlowDec's postfilter? Could the authors specify what benefits NDAC offers over DAC in this proposed two-stage setup?
>
> **Quote from Reviewer QBQt: “Contrary to adversarial training (a dominant approach in vocoder/codec) which requires domain specific expertise to stabilize training and tune the hyperparameters of multiple losses, FlowDec simplifies the trianing pipeline by elimiating the adversarial losses.“**
>
> In preliminary experiments, we tried using adversarial DAC as the underlying codec. The postfilter suppressed and enhanced some  artifacts from DAC, e.g. spurious high-frequency harmonics in speech. **However, DAC also sometimes fully lost signal details, in which case the postfilter did not convincingly resynthesize them.** Specifically, we found that DAC can suffer from spectral smoothing and smearing (harmonics spread across frequencies) - a problem that NDAC avoids. For example, see the 4-6 kHz bands in the middle row in our updated Fig. 14 (NDAC output added for comparison).
>
> **Some issues with adversarial training that we avoid are:**
>
> * **The need for highly handcrafted multi-discriminator losses and weights to avoid training instabilities, mode collapse, or divergence.** Our FM objective is simple and highly stable in our experience (no LR scheduler or gradient clipping).
> * **The ambiguity and lack of interpretability of adversarial training**: the discriminator makes black-box decisions that may not align with the goal but are difficult to inspect. Our FM objective minimizes the Wasserstein-2 distance to the data distribution, thus optimizing perceptual quality in the sense of [5*].
> * **The lack of detail modeling capability compared to diffusion-/flow-based methods**, as demonstrated in many diffusion-based SOTA image generation works in recent years. As shown in Fig. 7, FlowDec shows improved high-frequency modeling in some difficult cases where DAC generates noise-like structures.
>
> The authors of the concurrent work PeriodWave [4*] also discuss limitations of GAN-based methods. **We now discuss these issues more clearly in our updated section 3.3.**
>
> **Finally, we argue that it is more interpretable and theoretically better grounded to cleanly separate a postfilter-based approach into two modules**, see also a recent related arXiv submission on image restoration, https://arxiv.org/pdf/2410.00418, which offers additional theoretical arguments from this viewpoint.
>
> > The STFT settings seem unconventional. [...]
>
> We understand the reviewer’s concern. The computational cost of the STFT is indeed negligible overall.  When using 769 bins, the underlying U-Net NCSN++ network [3*] must be adapted, but doing so we experienced issues with upsampling padding artifacts. **We have based the STFT parameters on prior work [2*]** to avoid any such artifacts.

---

> > ### Author Response · Authors · 2024-11-21
> > **Mentioned references**
> >
> > [1*] Neil Zeghidour, Alejandro Luebs, Ahmed Omran, Jan Skoglund, and Marco Tagliasacchi. SoundStream: An end-to-end neural audio codec. IEEE/ACM Transactions on Audio, Speech, and Language Processing, 30:495–507, 2021.
> >
> > [2*] Julius Richter, Yi-Chiao Wu, Steven Krenn, Simon Welker, Bunlong Lay, Shinji Watanabe, Alexander Richard, and Timo Gerkmann. EARS: An anechoic fullband speech dataset benchmarked for speech enhancement and dereverberation. ISCA Interspeech, pp. 4873–4877, 2024.
> >
> > [3*] Yang Song, Jascha Sohl-Dickstein, Diederik P Kingma, Abhishek Kumar, Stefano Ermon, and Ben Poole. Score-based generative modeling through stochastic differential equations. International Conference on Learning Representations (ICLR), 2021.
> >
> > [4*] Lee, Sang-Hoon, Ha-Yeong Choi, and Seong-Whan Lee. "PeriodWave: Multi-Period Flow Matching for High-Fidelity Waveform Generation." arXiv preprint arXiv:2408.07547 (2024).
> >
> > [5*] Blau, Yochai, and Tomer Michaeli. "The perception-distortion tradeoff." Proceedings of the IEEE conference on computer vision and pattern recognition. 2018.

---

### Official Review · Reviewer_ofGs · 2024-11-02

**Soundness:** 3
**Presentation:** 4
**Contribution:** 3
**Rating:** 8
**Confidence:** 4

**Summary:**

This paper proposes FlowDec, a 48 kHz general audio codec with a flow-matching diffusion post-filter. FlowDec modifies the DAC audio codec with different loss functions, the stochastic post-filter, and frequency-dependent noise. Evaluations demonstrate strong improvements over prior post-filter and diffusion-based methods, but only small improvements over DAC.

**Strengths:**

Figures 2 and 3 are great visualizations for the diffusion dynamics and proposed improvements.

Good selection of baselines, including retraining baseline models to account for differences in parameter count.

The authors mention that they will be open-sourcing their code.

**Weaknesses:**

You should consolidate the six main contributions. I recommend at most three or four. Some of the contributions currently listed seem minor, or are a side effect of another contribution.

A primary issue with the proposed system is RTF. The logical alternative (DAC) is still orders of magnitude faster. I do still think this is an important paper in helping close the gap.

A minor note, but a 44.1 kHz sample rate is sufficient to prevent audible aliasing above 20 kHz. The other rationales given for using 48 kHz over 44.1 kHz are convincing enough for me.

**Questions:**

The NeurIPS 2023 oral by Kingma and Gao unifies under a common framework the diffusion variants of, e.g., flow matching and optimal transport. This could be useful, given that both are discussed in your work. You might also take a look at their Appendix J, which has some discussion of the frequency analysis of diffusion noise that may have some overlap with your proposed frequency-dependent noise.

Can we hear some audio examples? It is difficult to judge the results otherwise, and the MUSHRA test (unlike an AB or ABX test) doesn't provide sufficient granularity to discern statistical significance between, e.g., FlowDec and DAC.

---

> ### Author Response · Authors · 2024-11-21
>
> We would like to thank the reviewer for their positive evaluation of our work and are happy that our work on explanatory visualizations is appreciated. **We will indeed open-source the code as mentioned, along with pretrained model checkpoints.**
>
> >Can we hear some audio examples? It is difficult to judge the results otherwise, and the MUSHRA test (unlike an AB or ABX test) doesn't provide sufficient granularity to discern statistical significance between, e.g., FlowDec and DAC.
>
> **We now provide a demo page at https://flowdec2024.github.io/FlowDecSupplementary/**.
>
> >The NeurIPS 2023 oral by Kingma and Gao unifies under a common framework the diffusion variants of, e.g., flow matching and optimal transport. This could be useful, given that both are discussed in your work. You might also take a look at their Appendix J, which has some discussion of the frequency analysis of diffusion noise that may have some overlap with your proposed frequency-dependent noise.
>
> We thank the reviewer for this nice remark and useful reference. It is interesting to see that the OT-FM objective is equivalent to v-prediction with a cosine schedule. We have now added a citation to this work in our section on frequency-dependent noise (section 3.5). The data-based frequency-dependent construction we propose in section 3.5 is designed for audio signals, but could potentially be employed for images too.
>
> >You should consolidate the six main contributions. I recommend at most three or four. Some of the contributions currently listed seem minor, or are a side effect of another contribution.
>
> We thank the reviewer for this valuable suggestion. **We have now consolidated contributions** (1) and (2) which are both theoretical developments, and (3) and (4) which are both improvements over ScoreDec. We have further removed (6) to focus on the core ideas of our work. Please see the updated manuscript.
>
> >A primary issue with the proposed system is RTF. The logical alternative (DAC) is still orders of magnitude faster. I do still think this is an important paper in helping close the gap.
>
> We understand the reviewer’s concern regarding this point. **Our main proposal with this work is that postfiltering methods can be competitive with SOTA GAN-based methods while avoiding some of their problems**, see our response to reviewer VB3E, and also have qualitative advantages in some scenarios.
>
> **We expect that future works can address the slow RTF** with further improved and more highly domain-adapted postfilter network architectures, further reduction of the number of function evaluations, the consolidation or joint training of the initial decoder with the postfilter, or other techniques from recent literature such as band-parallel processing.
>
> >A minor note, but a 44.1 kHz sample rate is sufficient to prevent audible aliasing above 20 kHz. The other rationales given for using 48 kHz over 44.1 kHz are convincing enough for me.
>
> We appreciate this note and have removed our sentence on avoiding aliasing accordingly.

---

> > ### Comment · Reviewer_ofGs · 2024-11-23
> >
> > I appreciate the authors' response and the updates made to an already good paper. I am increasing my score from 6 to 8.

---

### Meta-Review · Area_Chair_hVXa · 2024-12-07

**Metareview:**

The paper introduces FlowDec, a neural audio codec that employs a two-stage approach: (1) an autoencoder with residual vector quantization, trained without adversarial loss; and (2) a postfilter that mitigates coding artifacts and enhances perceptual quality. FlowDec leverages conditional flow matching for signal enhancement, achieving notable improvements over previous score-based and flow-based models. The only major issue raised by reviewers concerns the comparison with additional baselines, which was addressed during the rebuttal.

**Additional Comments On Reviewer Discussion:**

The only major issues raised by reviewers concern the baselines. Reviewer VB3E highlighted the weakness of not exploring a GAN-based codec combined with the proposed postfilter. The authors responded by discussing the drawbacks of the GAN-based approach as the reason for not further improving it, which I consider as acceptable answer. Reviewer bRLZ suggested including other diffusion-based works for comparison. The authors addressed this by providing the comparisons during the rebuttal.

---

### Decision · Program_Chairs · 2025-01-22

Accept (Poster)